# Molecular basis of the pleiotropic effects by the antibiotic amikacin on the ribosome

Savannah M. Seely [1,6], Narayan P. Parajuli [2,6], Arindam De Tarafder [2], Xueliang Ge [2], Suparna Sanyal [2] ✉ & Matthieu G. Gagnon [1,3,4,5] ✉

Aminoglycosides are a class of antibiotics that bind to ribosomal RNA and exert pleiotropic effects on ribosome function. Amikacin, the semisynthetic derivative of kanamycin, is commonly used for treating severe infections with multidrug-resistant, aerobic Gram-negative bacteria. Amikacin carries the 4-amino-2-hydroxy butyrate (AHB) moiety at the $N^1$ amino group of the central 2-deoxystreptamine (2-DOS) ring, which may confer amikacin a unique ribosome inhibition profile. Here we use in vitro fast kinetics combined with X-ray crystallography and cryo-EM to dissect the mechanisms of ribosome inhibition by amikacin and the parent compound, kanamycin. Amikacin interferes with tRNA translocation, release factor-mediated peptidyl-tRNA hydrolysis, and ribosome recycling, traits attributed to the additional interactions amikacin makes with the decoding center. The binding site in the large ribosomal subunit proximal to the 3'-end of tRNA in the peptidyl (P) site lays the groundwork for rational design of amikacin derivatives with improved antibacterial properties.

Antibiotics are an important arsenal used to treat bacterial infections. The majority of antibiotics inhibit protein synthesis by targeting the ribosome, the molecular machine responsible for decoding messenger RNAs (mRNAs) and correspondingly incorporating incoming amino acids into the nascent polypeptide chain. Structural studies of the ribosome in complex with antibiotics revealed that translation inhibitors generally target functional centers of the prokaryotic ribosome, the decoding center in the small (30S) subunit, and the peptidyl transferase center as well as the nascent peptide exit tunnel in the large (50S) subunit[1]. Aminoglycosides are broad-spectrum bactericidal antibiotics used to treat a wide spectrum of infections caused by Gram-negative pathogenic bacteria. The most common clinically used aminoglycosides harbor the central 2-deoxystreptamine (2-DOS) ring joined by glycosidic linkages with amino sugars. They interfere with protein synthesis by targeting the region of 16S ribosomal RNA (rRNA) helix h44 forming the decoding center within the 30S subunit of the ribosome (Fig. 1a, inset 1). Binding of 2-DOS-containing aminoglycosides to the decoding center displaces two universally conserved nucleotides, A1492 and A1493, promoting their interaction with the minor groove of the codon-anticodon duplex in the aminoacyl (A) site of the ribosome. This "ready-to-accept" conformation of the decoding center promotes the binding of near-cognate aminoacyl-tRNAs, thereby inducing translational errors[2–9].

The nature of the 2-DOS core ring structure and types of chemical substitutions allow to broadly classify aminoglycosides into four subclasses: (i) no 2-DOS (e.g. streptomycin), (ii) mono-substituted 2-DOS (e.g. apramycin), (iii) 4,5-di-substituted 2-DOS (e.g. neomycin, paromomycin), and (iv) 4,6-di-substituted 2-DOS (e.g. gentamicin, kanamycin, tobramycin)[10]. Despite their similar chemical structures, aminoglycoside antibiotics exert pleiotropic effects on ribosome function. For instance, neomycin and paromomycin, both belonging to the 4,5-di-substituted subclass, promote 30S subunit rotation in the

[1]Department of Biochemistry and Molecular Biology, University of Texas Medical Branch, Galveston, TX 77555, USA. [2]Department of Cell and Molecular Biology, Biomedical Center, Uppsala University, SE-75124 Uppsala, Sweden. [3]Department of Microbiology and Immunology, University of Texas Medical Branch, Galveston, TX 77555, USA. [4]Sealy Center for Structural Biology and Molecular Biophysics, University of Texas Medical Branch, Galveston, TX 77555, USA. [5]Institute for Human Infections and Immunity, University of Texas Medical Branch, Galveston, TX 77555, USA. [6]These authors contributed equally: Savannah M. Seely, Narayan P. Parajuli. ✉e-mail: suparna.sanyal@icm.uu.se; magagnon@utmb.edu

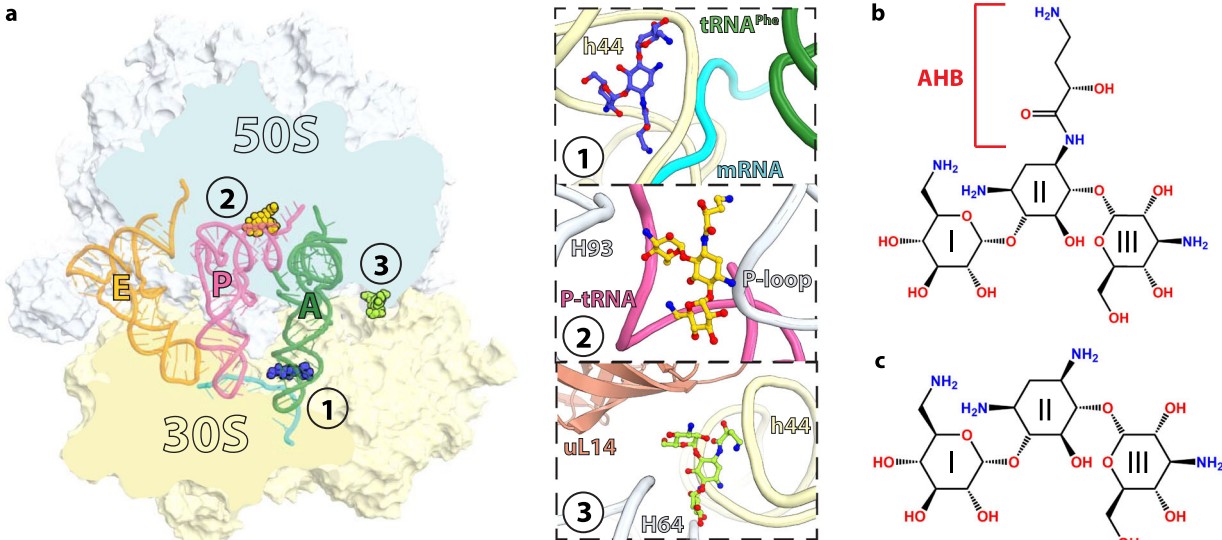

**Fig. 1 | Overview of the *Thermus thermophilus* 70S ribosome bound to amikacin. a** Overview of the 70S ribosome complexed with mRNA (cyan), tRNA^Phe in the aminoacyl (A) site (green), tRNAi^fMet in the peptidyl (P) site (pink), tRNA^Phe in the exit (E) site (orange), and AMK bound to three potentially relevant sites. (Inset 1) AMK bound near the decoding center in the small subunit, (inset 2) AMK bound in the large subunit P site, and (inset 3) AMK bound at inter-subunit bridge B5. Chemical structures of the antibiotics AMK with the amino-hydroxy butyrate (AHB) moiety at the N^1 position of the central 2-deoxystreptamine (2-DOS) ring II (**b**), and KAN (**c**).

opposite direction relative to the usual ratcheting motion observed during translation[11]. The unique inhibition profile of aminoglycosides may be due to additional binding sites in the ribosome other than the primary site in h44 of the 30S subunit. To this effect, neomycin interferes with ribosome recycling through its association with 23S rRNA helix H69, helping to maintain the inter-subunit bridge between H69 and h44[12]. Aminoglycosides also stabilize the classic state of tRNA binding, providing a basis by which they inhibit tRNA translocation[13].

The rise of antibiotic resistance has challenged the use of aminoglycosides in the therapy for bacterial infections. Yet, it spurred renewed interest in the legacy aminoglycosides and the development of novel semisynthetic aminoglycosides such as amikacin, arbekacin, and plazomicin[10]. Amikacin (AMK) is the most widely used semisynthetic aminoglycoside due to being refractory to the majority of aminoglycoside-modifying enzymes. AMK is synthesized by the addition of the 4-amino-2-hydroxy butyrate (AHB) group at the N^1 amino group of the 2-DOS moiety of kanamycin A (KAN) (Fig. 1b, c). Despite the common usage of AMK to treat a plethora of infections, the contribution of the AHB group to protein synthesis inhibition remains largely unknown. The crystal structure of an RNA mimic of the decoding center and the cryo-EM reconstruction of the 30S and 50S subunits of the *Acetinobacter baumannii* ribosome confirmed the binding of AMK to helix h44[9,14]. However, it is conceivable that the AHB group confers AMK a unique binding spectrum to functional ribosome complexes, which may account for the high inhibition potency of AMK on protein synthesis.

Here we use in vitro fast kinetics to show that, in addition to impeding mRNA translocation, AMK strongly inhibits release factor (RF)-mediated peptidyl-tRNA hydrolysis and interferes with ribosome recycling. While the inhibitory effects of AMK are attributed to its binding to the decoding center, our crystal and cryo-EM structures of the 70S ribosome show that one AMK molecule binds to the 50S subunit proximal to the CCA-end of the peptidyl (P)-site tRNA, the functional significance of which is unknown. However, AMK binding to this site provides an opportunity for the development of aminoglycoside derivatives with novel antibacterial properties. Our structural and kinetics data illustrate how AMK and KAN, two closely related aminoglycosides, distinctively inhibit ribosome function.

## Results

### Crystal structures of the ribosome bound to amikacin and kanamycin

The crystal structures of the *Thermus thermophilus* 70S ribosome in complex with mRNA, tRNAs, AMK or KAN were determined at ~2.9 Å resolution by molecular replacement using a high-resolution model of the 70S ribosome with the tRNA and mRNA ligands removed (see "Methods")[15]. The initial difference Fourier density maps calculated using the $F_{obs} - F_{calc}$ amplitudes revealed, as expected, clear unbiased density for AMK or KAN in the canonical aminoglycoside binding site in helix h44 of 16S rRNA (Fig. 1a, inset 1, Fig. 2a–c, Supplementary Fig. 1a–c). Additional unique sites are observed for AMK and KAN, providing insights into their ribosome binding modes (Fig. 1a, Supplementary Figs. 2 and 3). The peculiar location of two secondary AMK binding sites suggested that they may contribute to ribosome inhibition.

One secondary binding site for AMK is in the vicinity of the P-site tRNA acceptor stem 3′CCA-end and the P-loop of 23S rRNA (Fig. 1a inset 2, Fig. 3a, b). At this site, the majority of the interactions AMK makes with rRNA and tRNA are mediated by sugar-phosphate backbones. Rings II and III of AMK stack with the ribose of nucleotides G2253 (*Escherichia coli* nucleotide numbers are used throughout) of the P-loop and C74 of P-site tRNA (Fig. 3c, d). The AHB moiety and ring II form a surface that is chemically complementary with that of the universally conserved nucleotides G2252 and G2253 of the P-loop (Fig. 3d). The hydroxymethyl in ring III and the amine in ring II are within hydrogen-bonding distance of the non-bridging oxygen atoms of P-site tRNA A73 and C2254 of the P-loop, respectively, and the amine of the AHB group forms multiple H-bonding interactions with G2252 of the P-loop (Fig. 3d). The interactions mediated by the AHB group of AMK with the P-loop likely contribute to the binding of AMK to this site because KAN, which lacks the AHB moiety, does not bind to the P site of the 50S subunit.

The binding site of AMK in the P site of the 50S subunit is distinct from that of the translation inhibitors blasticidin S (BlaS) and bactobolin A (BacA)[16,17]. Contrary to BlaS and BacA, AMK does not interfere with the conformation of the CCA-end of P-tRNA (Fig. 3c). BlaS, a cytidine nucleoside analog, spatially replaces C75 of tRNA and

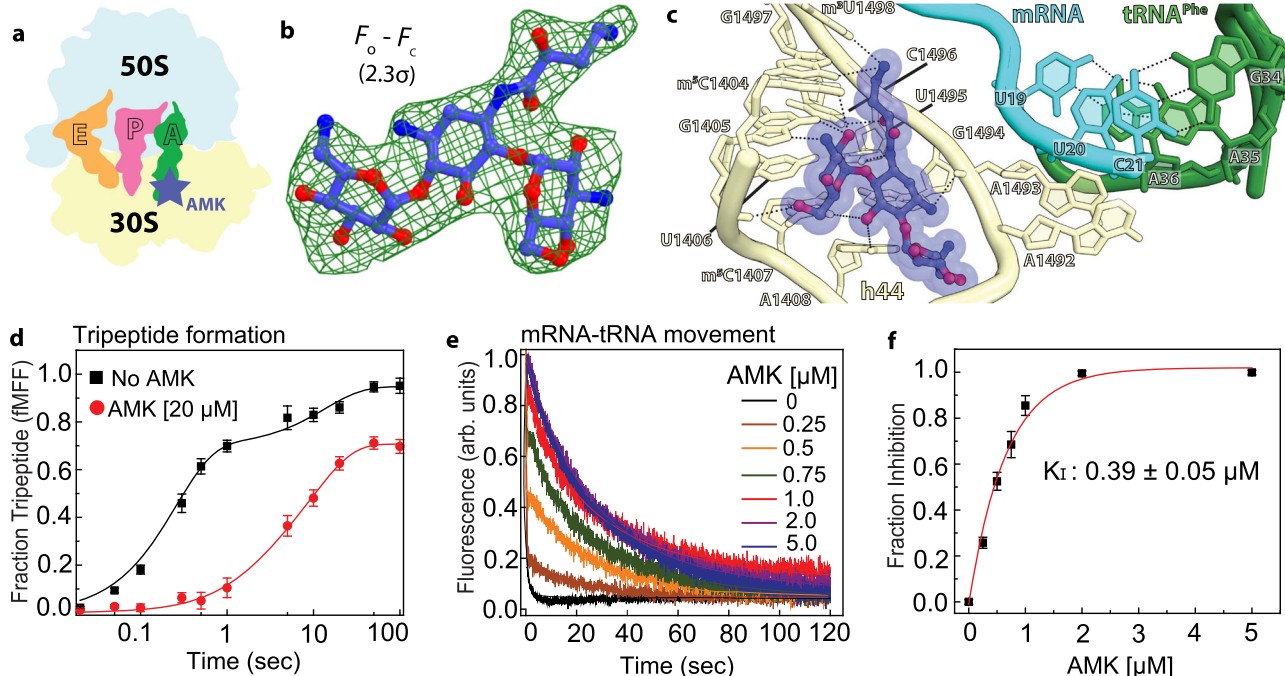

**Fig. 2 | Canonical binding site of amikacin near the decoding center. a** Simplified representation of the 70S ribosome with the AMK binding site indicated with the blue star. **b** The unbiased ($F_o - F_c$) difference electron density map of AMK bound near the decoding center is contoured at 2.3σ. **c** AMK binds within helix h44 of the decoding center where the AHB moiety forms three unique interactions. **d** Time courses of f[³H]Met-Phe-Phe tripeptide formation with EF-Tu ternary complex (TC) (5 μM) and EF-G (5 μM) in the absence (black) and presence of 20 μM AMK (red). Solid lines represent the double exponential fit of the data with SEM from $n = 3$ independent experiments. **e** Time evolution of fluorescence traces obtained for the

EF-G (5 μM) catalyzed movement of pyrene-labeled mRNA on 70S ribosomes (0.5 μM) in the presence of various concentrations (0-5 μM) of AMK. The inhibition of mRNA movement by AMK was estimated from amplitudes of the slow phase of fluorescence traces relative to the total transition (normalized to 1) indicative of inhibited fraction of the ribosomes. **f** The fraction of AMK-inhibited pre-TC plotted against AMK concentration. Data were fitted with hyperbolic function (solid line) and half-inhibitory concentration ($K_I$) of AMK on the inhibition of translocation was estimated from mid-point of transition. Experiments were conducted in triplicates and error bars indicate the SEM of data.

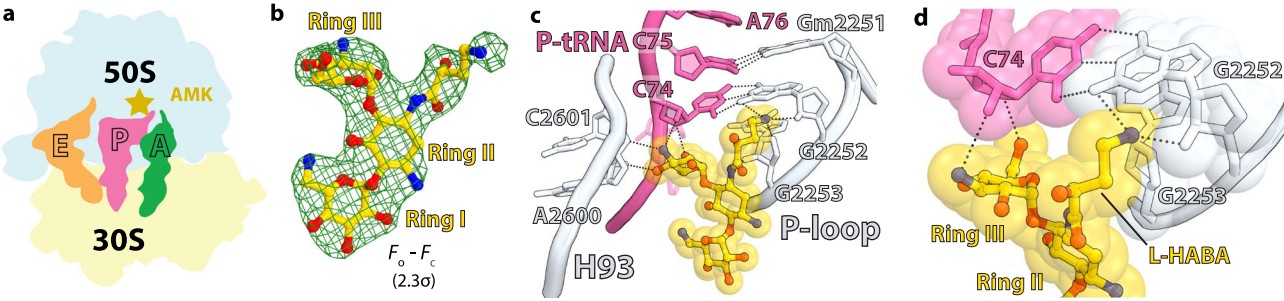

**Fig. 3 | Amikacin binding site in the large subunit proximal to the P-site tRNA. a** Cartoon representation of the 70S ribosome carrying three tRNAs with the AMK binding site indicated with the yellow star. **b** The unbiased ($F_o - F_c$) difference electron density map of AMK bound to the large subunit P site is contoured at 2.3σ.

**c** In the large subunit AMK (yellow) binds near the CCA-end of the P-site tRNA (pink), the conserved 23S rRNA P-loop (white), and helix H93 (white). **d** Interactions between AMK (yellow) and the Watson-Crick base pair G2252-C74 formed by the P-loop (white) and the P-tRNA CCA-end (pink).

displaces it toward the A site, which interferes with the accommodation of the catalytic domain 3 containing the GGQ loop in release factors (RFs), thereby inhibiting RF-mediated peptidyl-tRNA hydrolysis (Supplementary Fig. 4a)[16–20]. BacA also displaces the CCA-end of tRNA toward the A site, and correspondingly, is proposed to also inhibit RF-mediated peptide release (Supplementary Fig. 4b)[21].

In the ribosome, interactions between helices h44, H64, and ribosomal protein uL14 form bridge B5, the largest inter-subunit contact area[22,23]. The difference Fourier map of the *T. thermophilus* 70S-AMK complex revealed one AMK molecule bound between the 30S and 50S subunits at the center of bridge B5 (Fig. 1a inset 3, Fig. 4a–c). The inter-subunit contact surface area (-1185 Å²) is increased to -1430 Å² upon binding of AMK at bridge B5, suggesting that AMK strengthens

this inter-subunit contact and may stabilize the 70S ribosome (Supplementary Fig. 5a, b). At this site, AMK interacts with non-bridging oxygen atoms on the major groove side of helix h44 in the 30S subunit, with ribosomal protein uL14, and with nucleobases and the sugar-phosphate backbone facing the minor groove of helix H64 in the 50S subunit (Fig. 4c). While the 2-DOS ring II of AMK does not interact with the ribosome, ring I stacks with the ribose of C1988 in H64 (Fig. 4c). The amine and hydroxyl groups on one side of ring I are within hydrogen-bonding distance of non-bridging oxygen atoms of A1473 and G1474 in h44, and the hydroxyl groups on the other side of ring I interact with the 2′OH and exocyclic amino groups of G1987 in H64 (Fig. 4c). AMK reaches across the major groove of h44 with the amino group of the AHB side chain forming water-mediated hydrogen-

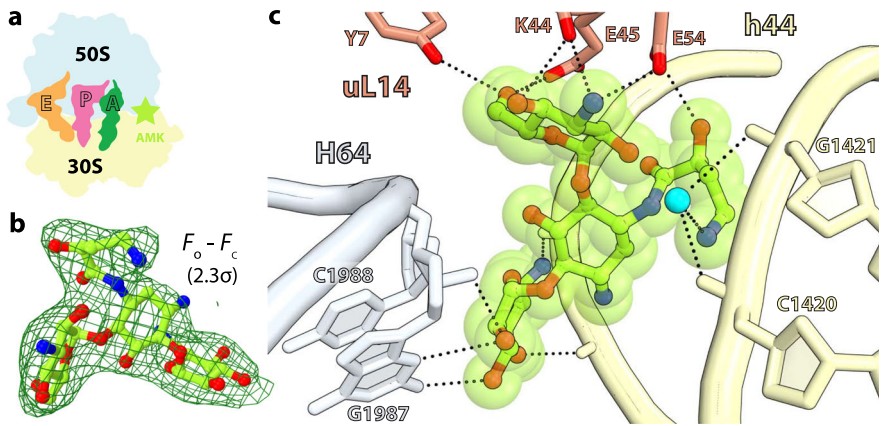

**Fig. 4 | Amikacin binding site at inter-subunit bridge B5. a** Simplified representation of the 70S ribosome with the AMK binding site indicated with the green star. **b** The unbiased ($F_o - F_c$) difference electron density map of AMK bound at the inter-subunit space is contoured at 2.3σ. **c** AMK interacts with elements of inter- subunit bridge B5, helix h44 of 16S rRNA, helix H64 of 23S rRNA, and ribosomal protein uL14. The amine group of the AHB group forms water-mediated (cyan) H-bonds with the phosphate backbone of h44.

bonding interactions with the phosphate oxygen atoms of C1420 and G1421 in h44 (Fig. 4c). The AHB hydroxyl group and the chemical moieties decorating ring III form multiple interactions with ribosomal protein uL14 residues Tyr7, Glu45, Glu54, and the main chain carbonyl oxygen of Lys44 (Fig. 4c). The observation that KAN does not bind to this site at the subunit interface of the *T. thermophilus* 70S ribosome indicates that the water-mediated interactions between the AHB moiety of AMK and helix h44 contribute to the binding of AMK at bridge B5.

## Amikacin interferes with mRNA translocation during peptide elongation

To assess the effects, if any, of AMK binding in the 50S subunit near the CCA-end of P-site tRNA, we first conducted kinetic experiments to estimate the rates of dipeptide (fMet-Phe) formation in the absence and presence of AMK. In these experiments, ternary complex (TC) of EF-Tu•GTP•Phe-tRNA[Phe] was rapidly mixed in a quench-flow with initiated ribosomes containing f[3H]Met-tRNA[fMet] in the P site and the accumulation of dipeptide (f[3H]Met-Phe) with time was monitored. The rates of dipeptide formation with AMK-free ribosomes ($56 \pm 8\,\text{s}^{-1}$) was virtually identical to that observed with AMK-bound ribosomes ($53 \pm 5\,\text{s}^{-1}$), indicating no effect of AMK on delivery and accommodation of the aminoacyl-tRNA and peptidyl transfer (Supplementary Fig. 6a). This agrees with the observation that AMK at this location does not interfere with the conformation of the CCA-end of the P-site tRNA (Fig. 3c, Supplementary Fig. 4a, b). Likewise, KAN also showed no effect on dipeptide formation (Supplementary Fig. 6a).

Within coordinate errors, we observe AMK bound to h44 in the same conformation as that previously reported from the structures of AMK bound to an RNA fragment of h44 and to the *Acinetobacter baumannii* 30S ribosomal subunit (Supplementary Fig. 7)[9,14]. Binding of AMK to h44 promotes the flipped-out conformation of the two universally conserved nucleotides A1492 and A1493 forming A-minor interactions with the mRNA-tRNA duplex in the A site (Fig. 1a inset 1, Fig. 2a–c, Supplementary Fig. 7). The aminoglycoside-induced conformation of the decoding center promotes amino acid misincorporations by facilitating the binding of near-cognate aminoacyl-tRNAs[24]. Binding of aminoglycosides to this site further exerts strong inhibition of tRNA translocation[13,24]. Single-molecule fluorescence resonance energy transfer (smFRET) studies reported that most aminoglycosides, including KAN, stabilize the classical state of tRNA binding and inhibit EF-G-catalyzed translocation[13,25,26].

Correspondingly, ablation of the primary binding site through the A1408G mutation in h44 increased the $K_I$ of EF-G-dependent translocation of neomycin and tobramycin by 100- and 25-fold, respectively[27]. Furthermore, the ability of neomycin to promote reverse translocation depends on its primary binding site in h44[27].

We next estimated the rates of tripeptide formation in the absence and presence of AMK. In these experiments, EF-Tu•GTP•Phe-tRNA[Phe] with EF-G•GTP were rapidly mixed with mRNA-programmed ribosomes containing f[3H]Met-tRNA[fMet] in the P site and time courses of tripeptide (fMet-Phe-Phe) formation were measured. While AMK showed no effect on dipeptide formation (Supplementary Fig. 6a), the rates for tripeptide formation dropped from $6.1 \pm 1.2\,\text{s}^{-1}$ to $1.1 \pm 0.5\,\text{s}^{-1}$ with the addition of 20 µM AMK (Fig. 2d). These results suggest that the binding of AMK to elongating ribosomes affects the stage between dipeptide and tripeptide formation i.e., ribosomal translocation. We then directly measured the kinetics of EF-G-catalyzed movement of mRNA-tRNA during ribosomal translocation using a fluorescence assay based on pyrene-labeled mRNA (Fig. 2e, f)[24,28]. Although this assay includes steps of aminoacyl-tRNA accommodation and peptidyl transfer prior to ribosomal translocation, the insensitivity of these processes toward AMK allows the precise estimation of its action on mRNA-tRNA movement. The fluorescence traces indicative of EF-G-catalyzed mRNA movement in the absence of AMK were nearly monophasic (95%), and the estimated rate of mRNA movement was $12.4 \pm 2\,\text{s}^{-1}$ (Fig. 2e). However, upon pre-incubation with AMK, biphasic fluorescence traces were observed (Fig. 2e). Interestingly, the mean-time ($96 \pm 12$ mSec) of the fast phase of fluorescence transition was comparable to that in the absence of AMK ($83 \pm 9$ mSec), probably reflecting translocation on AMK-free ribosomes. Increased concentrations of AMK yielded a more predominant slow phase, however with virtually identical mean times ($-18.3 \pm 1.7$ Sec), indicative of increased binding of AMK and delayed translocation on AMK-bound ribosomes (Fig. 2e). From the amplitudes of the slow phase, we then estimated the fraction of AMK-inhibited ribosomes prior to translocation and plotted them against each concentration of AMK (Fig. 2f)[24]. The fraction of inhibited ribosomes increased hyperbolically and reached its half-maximal value ($K_I$) at 0.39 µM AMK, identical to that of arbekacin (ABK) (~0.4 µM)[24], a similar AHB-containing semisynthetic aminoglycoside. Nevertheless, similar to ABK[24], all the fluorescence traces recorded in the presence of AMK reached the same basal level verifying the completion of mRNA translocation in the presence of AMK. By comparison, KAN lacking the AHB side chain shows much weaker inhibition of translocation with $K_I \sim 0.8$ µM (Supplementary Fig. 6d). The twofold higher $K_I$ of KAN than AMK can be attributed to

the fewer interactions of KAN with the decoding center due to the absence of the AHB group, which altogether reduces its affinity for the primary aminoglycoside binding site on the ribosome. The AHB group of AMK forms three additional hydrogen bonds with nucleotides C1496, G1497, and m³U1498 in h44 (Fig. 2c). These interactions are unique to AMK, and likely to ABK as well, and may rigidify the top of h44 which is known to bend by ~8 Å toward the P site during EF-G-mediated tRNA translocation[29], providing a plausible explanation for the similar inhibitory effect of AMK and ABK on the movement of mRNA and tRNAs.

## Amikacin inhibits release factor-mediated peptide release

Aminoglycosides have been reported to inhibit RF-mediated peptide release[30,31]. Recently ABK, which also harbors the AHB group, was shown to impair peptide release[24]. We therefore asked whether AMK interferes with RF-mediated peptidyl-tRNA hydrolysis. To this end, we prepared pre-termination ribosome complexes (pre-TC) harboring the (BOP)•Met-Phe-Leu tripeptide attached to tRNA^Leu in the P site and a stop codon (UAA) in the A site, and rapidly mixed in a stopped-flow instrument with a RF mixture containing an excess of RF2. The resulting time courses of fluorescence transition due to the release of the (BOP)•Met-Phe-Leu tripeptide followed a nearly monophasic curve indicating a single-round of peptide release (Fig. 5a). The apparent rate of peptide release from the AMK-free pre-TC was $7.1 \pm 0.8\,s^{-1}$. Upon addition of AMK to the pre-TC, we observed remarkable inhibition of peptide release. The rates estimated from the predominant fast phase (> 99%) were similar without or with different concentrations of AMK. However, the fluorescence amplitudes decreased with increasing AMK concentration indicating that AMK-bound ribosomes are practically incapable of peptide release (Fig. 5a). The fraction of inhibited pre-TCs determined by the fractional loss of fluorescence amplitudes increased hyperbolically with the concentration of AMK, giving a half-maximal inhibitory concentration ($K_I$) of $0.15 \pm 0.02\,\mu M$ (Fig. 5b). The antibiotic KAN, the parent compound of AMK, has no measurable effect on the termination step of translation (Supplementary Fig. 6b). Noteworthy is that ABK inhibits peptide release with a much higher $K_I$ value (0.6 μM for RF1 and 0.5 μM for RF2)[24]. These results show that AMK is probably the most potent inhibitor of RF-mediated peptide release among all known aminoglycosides.

## Amikacin inhibits recycling of the ribosome by EF-G and RRF

During ribosome recycling, the inter-subunit bridges are dissolved and the 70S ribosome dissociates into individual 30S and 50S subunits. This process is catalyzed by the coordinated action of two translation factors in bacteria, the ribosome recycling factor (RRF) and elongation factor-G (EF-G)[32–39]. We measured the effects of AMK on post-termination ribosome splitting into subunits by RRF and EF-G. We prepared a post-termination 70S ribosome complex (post-TC) programmed with a deacylated-tRNA in the P site and subjected it to dissociation into subunits upon mixing with RRF, EF-G, and IF3 in a stopped-flow instrument. The time course of ribosome splitting was then monitored by following the decrease in Rayleigh light scattering in the absence and presence of various concentrations of AMK (Fig. 5c). The traces were fitted with a double exponential function and mean times and amplitudes of the fast phases indicating single-round ribosome splitting were determined. In the absence of AMK, the meantime of ribosome splitting was $205 \pm 8.6\,mSec$, which increased to $574 \pm 14\,mSec$ upon the addition of 10 μM AMK (Fig. 5c). Notably, the amplitude of the fast phase decreased with the increase of AMK. The fraction inhibited was estimated by the fractional loss of the amplitude of the fast phase, which increased hyperbolically and saturated with 10 μM AMK (Fig. 5d). The half-maximal concentration of AMK for ribosome recycling ($K_I$) was estimated from the transition mid-point as $7.5 \pm 0.8\,\mu M$ (Fig. 5d). In comparison, KAN has only a marginal effect on ribosome recycling (Supplementary Fig. 6c). Also,

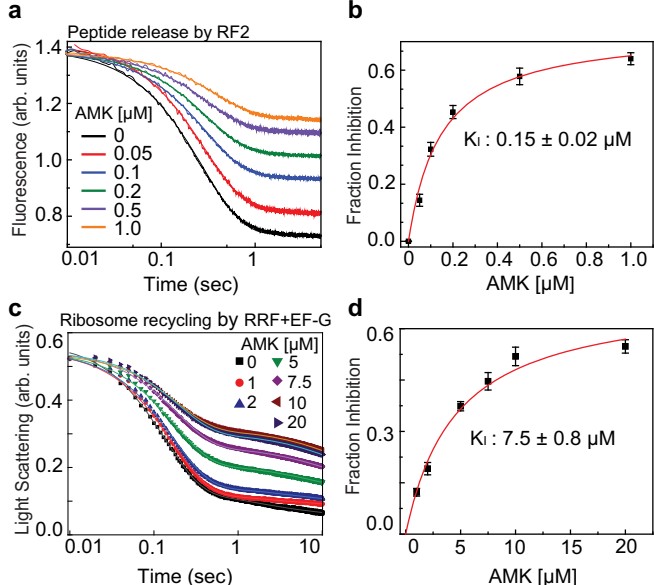

**Fig. 5 | Effects of amikacin on the kinetics of peptide release and ribosome recycling. a** Time courses of BOP-Met-Phe-Leu release from the P site of the ribosomes in pre-TC (0.1 μM) upon mixing with RF2 (1 μM) in the presence of various concentrations of AMK (0-1 μM). The near monophasic curves are fitted with double exponential function (solid lines) and the rates and amplitudes of the predominant fast phase (> 99%) were determined. The fraction inhibited was estimated from the fractional loss in fluorescence amplitude for a given AMK concentration considering the total amplitude of fluorescence transition (without AMK) as 1. **b** Fraction inhibition of RF2-mediated peptide release as the function of increasing concentrations of AMK. Solid line is the hyperbolic fit of data from which half-maximal inhibitory concentration ($K_I$) of AMK for peptide release was estimated. **c** Time traces for Rayleigh light scattering upon splitting of post-TC ribosomes (0.5 μM) into subunits by the concerted action of RRF (20 μM) and EF-G (10 μM) in the presence of various concentrations of AMK (0–20 μM). The scattering traces were fitted with double exponential function and the rates and amplitudes of both the fast and slow phases were determined. **d** Fraction inhibition of RRF and EF-G-mediated ribosome splitting was estimated from the fractional loss of the amplitude of the fast phase considering amplitude of the entire transition without AMK as 1. The solid line represents the hyperbolic fit of the fraction inhibition plotted against AMK concentration from which the half-maximal concentration ($K_I$) of AMK to inhibit ribosome recycling was estimated. Experiments were conducted in triplicates, data were fitted in Origin(Pro), Version 2016 (OriginLab Corp.), and error bars indicate the SEM of data.

the reported $K_I$ value for the similar aminoglycoside ABK in ribosome recycling is 30 μM[24], suggesting that AMK is the most potent aminoglycoside inhibitor of ribosome splitting.

## Cryo-EM structure of the *E. coli* ribosome bound to amikacin

The differential inhibition of translation by AMK and KAN may originate from the secondary binding sites observed in the *T. thermophilus* ribosome. However, the presence of secondary sites raises the legitimate concern that they may not be populated in the *E. coli* ribosomes at the drug concentrations used in the kinetics experiments. The crystals of *T. thermophilus* ribosomes were transferred into cryo-protection buffer solutions containing a high concentration (100 μM) of AMK or KAN, and then harvested and frozen. Likewise, the concentration of the ribosomes inside the crystals is ~0.2 mM, thereby representing non-physiological conditions.

To elucidate whether AMK binds to the P site in the 50S subunit and at inter-subunit bridge B5 of the *E. coli* ribosomes, we assembled a complex with mRNA, tRNAs, and a 20-fold molar excess of AMK, closely mimicking the conditions used in the kinetics assays, and subjected the sample to cryo-EM (Supplementary Fig. 8). In this structure,

refined to a nominal resolution of 2.9 Å (Supplementary Fig. 9), AMK is bound near the decoding center in h44 and to the P site of the 50S subunit proximal to the tRNA as seen in the *T. thermophilus* ribosome (Supplementary Fig. 10a, c). This observation agrees with AMK forming backbone-mediated interactions with the 23S rRNA and the CCA-end of the P-site tRNA, suggesting this binding site is universal. However, there is no density for AMK at bridge B5 in the *E. coli* ribosome, corroborating the non-conserved nature of Tyr7 and Glu54 in ribosomal protein uL14 (Supplementary Fig. 11), residues that interact with AMK at this site in the *T. thermophilus* ribosome (Fig. 4c). Increasing the concentration of AMK to 100 μM, the same as we used to soak the *T. thermophilus* ribosome crystals, did not populate this site (not shown), indicating that it is species-specific. Therefore, the inhibition of recycling of the *E. coli* ribosomes is not mediated through the binding of AMK at the subunit interface.

Two additional AMK molecules are observed in the *T. thermophilus* 70S ribosome, one in the 30S subunit and one in the 50S subunit (Supplementary Fig. 2a). In the small subunit, AMK interacts with helix h7 in the body domain of the 30S subunit (Supplementary Fig. 2b–d). In the large subunit, AMK contacts helices H18 in domain I and H28 in domain II of the 23S rRNA. At this site, AMK also interacts with Arg168 of ribosomal protein uL4 (Supplementary Fig. 2e–g). In the *E. coli* 70S ribosome, there is no density for AMK at these two sites, presumably indicating low-affinity binding sites. We note that the shorter helix H28 in the *E. coli* 50S subunit ablates one side of the drug binding pocket, suggesting that it may be a species-specific site.

In addition to binding to h44 near the decoding center, KAN binds to three secondary sites in the *T. thermophilus* 50S subunit (Supplementary Fig. 3a). One KAN molecule interacts with helix H88 located in domain V of 23S rRNA (Supplementary Fig. 3b–d), the other KAN contacts helices H40 and H42 in domain II (Supplementary Fig. 3e–g), and the third KAN is found at the base of the A-site finger helix H38, interacting with helix H85 and with Lys8 of protein uL16 (Supplementary Fig. 3h–j). While these secondary binding sites are not likely to be physiologically relevant and probably the result from the high concentration of drug used during crystallization, they nevertheless enrich the catalog of interactions between small molecules and RNA.

## A1408G ribosomes are resistant to amikacin

We used *E. coli* ribosomes carrying the A1408G mutation to explore the functional significance of the AMK binding site in the 50S subunit proximal to the CCA-end of the P-site tRNA. In the presence of 100 μM AMK or KAN, the rate of the EF-G-catalyzed movement of mRNA during ribosomal translocation is unchanged (Supplementary Fig. 12a), indicating that AMK bound near the peptidyl transferase center does not interfere with tRNA translocation, consistent with a previous report showing that BlaS, which also binds in the vicinity of AMK in the 50S subunit P site, did not affect the apparent rate of EF-G-catalyzed translocation[40]. Likewise, RF2-mediated peptide release from the A1408G mutant ribosomes is unaltered by AMK (Supplementary Fig. 12b), again showing that the mutant ribosomes are fully resistant to AMK.

The minimum inhibitory concentrations (MICs) of AMK and KAN for wild-type and A1408G mutant strain of *E. coli* SQ171 were determined using the standard broth microdilution method. Our results indicate that the MICs of KAN for wild-type (8 μg/mL) and A1408G (≥256 μg/mL) were high compared to the MICs of AMK for wild-type (1 μg/mL) and A1408G (≥16 μg/mL) (Supplementary Table 1). The variation is possibly due to the difference in the affinity of these two drugs for the primary binding site in h44 near the decoding center within the 30S subunit.

The binding of RF2 to the stop codon in the A site causes nucleotide A1493 to stack inside h44 in a position that is not compatible with bound AMK (Supplementary Fig. 13a)[41], possibly explaining the inhibition of RF2-mediated peptidyl-tRNA hydrolysis. Similarly,

RRF and EF-G bound to the ribosome favor the intra-h44 stacking of A1492, which would in turn collide with AMK (Supplementary Fig. 13b)[33]. Taken together, our findings are consistent with the primary binding site in h44 for AMK being responsible for the inhibition of mRNA translocation, RF-mediated peptide release, and ribosome recycling.

## Discussion

Aminoglycoside antibiotics are known to bind to secondary sites in the ribosome. For instance, neomycin and tobramycin associate with helix H69 in the 50S subunit, in addition to the canonical site in h44 of the 30S subunit[12]. It is therefore challenging to disentangle the physiological role and the contribution of each binding site to ribosome inhibition. One approach is to ablate the canonical aminoglycoside binding site in h44 with the A1408G mutation, which then allows to assess the effects of the other sites in translation inhibition. This strategy was previously used to decipher the mechanism by which tobramycin and neomycin inhibit ribosome recycling[27]. Interestingly, the inhibition of recycling of the A1408G ribosomes by neomycin and tobramycin is virtually the same as with the wild-type ribosomes, showing that the binding of these aminoglycosides to H69 is likely responsible for this inhibition.

In this work, we report that AMK and KAN exhibit unique translation inhibition profiles. These results, together with the reported role of the secondary site in H69 for neomycin and tobramycin on ribosome inhibition[27], prompted us to systematically probe the function of the secondary binding sites of AMK identified in the structure of the *T. thermophilus* ribosome (Fig. 1, Supplementary Fig. 2). Initially, the data showing that KAN is a less efficient ribosome inhibitor than AMK seemed to correlate with the presence of secondary binding sites for AMK that may be of physiological relevance. We employed a two-pronged approach to elucidate the role of the secondary binding sites for AMK in the P site of the 50S subunit and at inter-subunit bridge B5.

We first analyzed the conservation of the residues in ribosomal protein uL14 at bridge B5 that interact with AMK. The most striking difference between *T. thermophilus* and *E. coli* is at position 54 (Supplementary Fig. 11). In *T. thermophilus*, Glu54 forms a hydrogen bond with the amine of ring III in AMK (Fig. 4c). In other representative bacteria, a basic residue (K or R) occupies position 54 (Supplementary Fig. 11), which may alter the binding site for AMK. This observation suggested that AMK may not bind to bridge B5 in the *E. coli* ribosomes, which we used in the kinetics experiments.

We addressed this caveat by using cryo-EM to visualize a complex between AMK and a functional *E. coli* ribosome bound to mRNA and tRNAs. Here, the concentrations of ribosomes and AMK closely mimic those used in the kinetics assays. The density map of the reconstruction unambiguously shows that AMK is not bound at bridge B5 in the *E. coli* ribosomes, ruling out any inhibitory effect mediated by this secondary site. In this EM map, the clear density for AMK in the P site of the 50S subunit near the CCA-end of tRNA (Supplementary Fig. 10c), and the RNA phosphate backbone-mediated interactions with AMK at this location, suggested that this binding site is likely universal. However, three-dimensional (3D) variability analysis focused on the AMK binding site in the 50S subunit revealed that less than 30% of the ribosomes contained clear density for the drug at this site (Supplementary Fig. 8), suggesting that AMK has a lower affinity for the P site of the 50S subunit than for the canonical site in h44. This observation is also consistent with the 16-fold higher MIC of AMK for the *E. coli* strain expressing A1408G mutant ribosomes (Supplementary Table 1).

To further assess the relevance of the binding site in the 50S subunit near the P-site tRNA, we used the A1408G ribosomes to perform kinetics of mRNA translocation and RF2-mediated peptide release (Supplementary Fig. 12). These assays conclusively show no inhibition by AMK, indicating that the binding site proximal to the tRNA in the 50S subunit does not affect ribosome function. While the

inhibition of ribosome recycling by neomycin and tobramycin could be attributed to the binding site in H69[27], our results associate all the ribosomal inhibitory effects of AMK to the canonical binding site in h44.

It is intriguing that mRNA translocation proceeds to completion in the presence of 5 μM AMK (Fig. 2e), while AMK-bound ribosomes are incapable of peptide release and recycling at the highest concentration of AMK (Fig. 5a, c). The selection of aminoacyl-tRNAs by the ribosome involves the monitoring nucleotides A1492 and A1493 in h44, which probe the geometry of the minor groove of the mRNA-tRNA anticodon helix. For the A-site tRNA to translocate to the P site, the monitoring bases disengage from the tRNA·mRNA complex, a process that is facilitated by EF-G[42,43]. It is possible for A1492 and A1493 to remain unstacked and not interfere with bound AMK in h44, which would allow mRNA translocation to proceed despite the presence of AMK. However, the binding of RF2 and RRF to the ribosome cause nucleotide rearrangements in the decoding center that are not compatible with AMK bound in h44 (Supplementary Fig. 13a, b).

In the structure of the pre-recycling non-rotated 70S ribosome complex with RRF and EF-G, domain II of RRF remodels the tip of H69 and the decoding center in the 30S subunit[33]. In this complex, A1492 stacks inside h44 and is within interaction distance of A1408. In this conformation, A1492 is not compatible with AMK bound to its primary site (Supplementary Fig. 13b). The competition between AMK and A1492 for the same site seemingly explains why, over the time course of the experiment, the ribosome recycling reaction does not reach completion (Fig. 5c). Similarly, during translation termination, binding of RF2 to the stop codon triggers rearrangements of A1493, and the stacking of A1493 within h44 is not compatible with bound AMK (Supplementary Fig. 13a)[41]. The reaction of RF2-mediated peptide release does not reach completion, presumably because ribosomes that are bound to AMK do not simultaneously bind to RF2 (Fig. 5a). The higher affinity of AMK for h44, relative to KAN, explains their different inhibition profiles. It is likely that the remodeling of the ribosome decoding center upon binding of RRF and RF2 promotes dissociation of KAN from the canonical site. The effects appear to be similar to the inhibition of the RF-catalyzed peptide release by neomycin[30] and paromomycin[31]. In agreement with this premise, binding of RF1 to the ribosome promoted dissociation of paromomycin from h44[31].

Despite the apparent non-physiological relevance of the AMK binding site in the P site of the 50S subunit, it nevertheless provides valuable information for future development and repurposing of old antibiotics. This approach represents a promising strategy to circumvent the spread of resistance to the drugs currently in use[44,45]. The adjacency of the AMK binding site near the tRNA in the large ribosomal subunit to the old antibiotics BlaS and BacA represents an opportunity to generate chimeric molecules that may have improved antibacterial properties and activity. A similar strategy was used to generate radezolid[46], a chimeric molecule between linezolid and sparsomycin based on their overlapping binding sites within the peptidyl transferase center of the 50S subunit[19,47]. Radezolid has a higher affinity for the ribosome than linezolid, which provided improved antibacterial activity against various Gram-positive bacteria[46]. The nucleoside cytidine analog BlaS, and the antitumor antibiotic BacA, are toxic to prokaryotic and mammalian cells[16,48–50]. Derivatives of BacA and BlaS, based on their structures bound to the ribosome[18,20,21], are explored to improve drug-like properties and circumvent the inhibitory activity toward eukaryotic cells while retaining potency against prokaryotic ribosomes[51–55]. Their adjacency to AMK may provide strategies to solve this challenging issue.

Collectively, our findings illustrate how two closely related antibiotics, AMK and KAN, exhibit pleiotropic ribosome inhibition activities. It is remarkable how the AHB chemical group increases the efficiency of translation inhibition by AMK. It will be worth exploring further modifications of aminoglycosides, and in particular of AMK,

which could increase the binding affinity to the P site of the 50S subunit. The ribosome inhibition activity of such AMK variants could be potentiated by impeding the movement of the acceptor domain of the P-site tRNA as it transits to the E site.

## Methods

### Purification of 70S ribosomes, mRNAs, initiator tRNA, and tRNA^Phe

*Thermus thermophilus* 70S ribosomes were purified as described previously[56] and resuspended in buffer containing 5 mM HEPES-KOH, pH 7.5, 50 mM KCl, 10 mM NH$_4$Cl, 10 mM Mg(CH3COO)$_2$, and 1 mM β-mercaptoethanol (β-ME), at a concentration of approximately 800 A$_{260}$/mL, flash-frozen in liquid nitrogen, and stored in small aliquots at −80 °C until use in crystallization experiments. The *E. coli* strain SQ171 containing ribosomes with the A1408G mutation in the *rrsB* gene was kindly provided by M. Johansson's laboratory[57]. Tight coupled *E. coli* 70S ribosomes (MRE600 and SQ171-A1408G) were purified following standard procedures[58]. The 24-mer XR7 mRNA, 5′-GCC *AAG GAG G*UA AAA **AUG** UUC UAA-3′ with strong Shine-Dalgarno sequence (AAG-GAGG) (in italics), AUG start codon (bold) followed by the UUC (Phe) (underlined) and UAA (stop) codons was chemically synthesized by Integrated DNA Technologies (Coralville, IA). Other XR7-mRNAs with ORF sequence AUGUUCUUCUAA (Met-Phe-Phe-stop) and AUGUUC-CUGUAA (Met-Phe-Leu-stop) were transcribed in vitro and prepared as in ref. 59. Pyrene-labeled mRNA+10 (sequence 5′-UAACAAU *AAGGAGG*UAUUAA**AUGUUCCUGU**-3′-pyrene) coding for Met-Phe-Leu were from IBA-biosciences, Germany[28]. The *E. coli* tRNA^Phe and tRNA$_i$^fMet were expressed and purified as previously described[60]. Nucleotides (ATP, UTP, CTP and GTP) were from Cytiva. All other analytical grade chemicals including amikacin sulfate and kanamycin sulfate were from Sigma-Aldrich (cat# K-1876 and A-2324, respectively).

All in vitro kinetic experiments were carried out in HEPES polymix buffer (pH 7.5) (5 mM HEPES, 95 mM KCl, 5 mM NH$_4$Cl, 5 mM Mg(OAc)$_2$, 8 mM putrescine, 0.5 mM CaCl$_2$, 1 mM spermidine and 1 mM 1,4-dithioerythritol) at 37 °C with energy regeneration components (1 mM ATP, 1 mM GTP, 10 mM phosphoenolpyruvate (PEP), 1 μg/ml pyruvate kinase and 0.1 μg/ml myokinase) ensuring cellular free Mg$^{2+}$ concentration (~2 mM)[24].

### Dipeptide and tripeptide formation experiments

Two mixtures, initiation mix (IM) and elongation mix (EM), were prepared in the HEPES polymix buffer. IM contained 70S ribosomes (0.5 μM), f[$^3$H]Met-tRNA$_i$^fMet (0.55 μM), mRNA Met-Phe-Phe (1 μM), IF1 (0.5 μM), IF2 (0.5 μM) and IF3 (0.5 μM) and EM was comprised of tRNA^Phe (5 μM), EF-Tu (5 μM), EF-Ts (2 μM), Phe (200 μM), and Phe-tRNA^Phe synthetase (1.5 units/μl). For tripeptide experiments, EM was supplemented with EF-G (5 μM). To test the effects of AMK and KAN, 20 μM of each drug was added to both IM and EM. Both mixes were incubated for 15 min at 37 °C. After incubation the IM and EM were rapidly mixed in a quench-flow instrument (RQF-3; KinTek Corp., USA) and the reactions were quenched with formic acid (17% final) at definite time intervals. Samples were processed as described earlier[24] and the relative amounts of f[$^3$H]Met, f[$^3$H]Met-Phe, and f[$^3$H]Met-Phe-Phe in the supernatant were separated using a reverse-phase chromatography column (C-18, Merck) connected to a Waters HPLC system coupled with the in-line ß-RAM radioactive detector. The rates of dipeptide and tripeptide in the absence and presence of AMK and KAN were estimated by fitting the data to a single exponential function using Origin(Pro), Version 2016 (OriginLab Corp., Northampton, MA, USA). Experiments were conducted in triplicates and average data was plotted with SEM.

### Pyrene-mRNA based assay for ribosomal translocation

Initiation mix (IM) was prepared essentially in a similar way as in the case of dipeptide experiments, except that XR7-mRNA in IM was replaced with 3′ pyrene-labeled mRNA+10 (coding for Met-Phe-Leu)[28].

AMK (0–5 µM) was added to IM as indicated. EM was prepared as in the case for tripeptide experiments. Both mixes were incubated for 15 min at 37 °C. Equal volumes of IM and EM were rapidly mixed in a stopped-flow instrument (µSFM BioLogic) at 37 °C and the fluorescence transition was monitored using 360-nm long-pass filter (Comar Optics Ltd.) after exciting at 343 nm. The resultant fluorescence traces were fitted with a double exponential function using Origin(Pro) 2016. Experiments were conducted in triplicates.

## Measurement of RF-mediated peptide release

Pre-termination ribosome complex (Pre-TC) containing BODIPY™ (BOP)•Met-Phe-Leu-tRNA$^{Leu}$ tripeptide in the P site and a stop codon (UAA) in the A site was prepared in HEPES polymix buffer (pH 7.5)[24,61]. Equal volumes of pre-incubated pre-TC (0.1 µM) and RF mixture containing RF2 (1 µM) were rapidly mixed in a stopped-flow instrument (µSFM BioLogic) at 37 °C. To assess the effect of AMK and KAN on peptide release, indicated concentration of each drug was added to both mixes. The release of BOP-Met-Phe-Leu tripeptide was followed by monitoring the decrease in BOP fluorescence (excitation: 575 nm) with a cutoff filter of 590 nm. The fluorescence traces were fitted with a double exponential function using Origin Pro 2016 and the rates and amplitudes of the predominant fast phase were estimated[61]. The fraction of the ribosomes inhibited with a given concentration of AMK was estimated by subtraction of the amplitude of the fluorescence curves with AMK from the one without AMK, divided by the total fluorescence change (without any drug). The fraction inhibition was plotted as a function of AMK concentration and fitted with hyperbolic equation to determine the half-maximal inhibitory concentration ($K_I$). Experiments were conducted in triplicates and average data was plotted.

## Ribosome recycling

Post-termination ribosome complex (post-TC), with an empty A site and deacylated tRNA in the P site, was prepared by mixing 70S ribosomes (0.5 µM) with XR7-mRNA (Met-Phe-Leu) (1 µM) and deacylated tRNA$^{Leu}$ (1 µM) in HEPES−polymix buffer. A factor mix (FM) containing RRF (20 µM), EF-G (10 µM), and IF3 (1 µM) was prepared. AMK (0–20 µM) or KAN (0–100 µM) was added to both post-TC and FM. Both mixes were incubated at 37 °C for 5 min. Equal volumes of post-TC and FM were rapidly mixed in a stopped-flow instrument (µSFM BioLogic) and the splitting of post-TC into subunits was monitored as a decrease in Rayleigh light scattering at 365 nm[62]. The rate of post-TC dissociation was estimated by fitting the data with the double exponential equation in Origin Pro 2016. The rates and amplitudes of the fast phases were determined. The fraction of the inhibited ribosomes was estimated by subtraction of the amplitude of the fast phase with AMK from the one without AMK, divided by total amplitude change (without any drug). The half-maximal inhibitory concentration ($K_I$) was estimated by plotting fraction inhibition against AMK concentration and fitting the data with hyperbolic function using Origin Pro 2016.

## Minimum inhibitory concentration (MIC) measurement

The minimum inhibitory concentrations (MICs) of AMK and KAN were determined by broth microdilution (BMD) method following Clinical and Laboratory Standard Institute (CLSI) guidelines for aminoglycosides[63]. Briefly, twofold serial dilutions of AMK and KAN were prepared in cation-adjusted Mueller Hinton broth (CA-MHBII) corresponding to the concentrations ranging from 0.25 to 256 µg/mL and added to the 96-well (12 × 8) round-bottomed microtiter plate. A control well containing only media without any antibiotic served as growth control. Bacterial suspensions equivalent to $5 \times 10^5$ CFU/mL (either with WT or A1408G mutant) prepared from a single colony of each strain from a freshly streaked agar plate into CA-MHBII, were added to the wells containing various AMK and KAN concentrations. The microtiter plates were incubated at 37 °C for 16 to 18 h and MIC was estimated as the lowest concentration of the AMK or KAN that

prevented the visible growth of bacteria. The results were interpreted according to the susceptibility breakpoints for AMK (susceptible ≤4 µg/mL; resistant ≥16 µg/mL) and KAN (susceptible ≤16 µg/mL; resistant ≥64 µg/mL) in CLSI guidelines[63]. E. coli ATCC 25922 was used as a reference quality control strain in all experiments. Experiments were conducted in triplicates (Supplementary Table 1).

## X-ray crystallographic structure determination

The ribosome complex was formed as previously reported with modifications[33]. The ribosomes were incubated with 8 µM 24-MF mRNA in buffer containing 5 mM HEPES-KOH pH 7.5, 10 mM Mg(CH3COO)$_2$, 50 mM KCl, 10 mM NH$_4$Cl, and 6 mM β-ME at 55 °C for 5 min. The tRNA$^{Phe}$ and tRNA$_i^{Met}$ were added to a final concentration of 20 µM and 8 µM, respectively, and the complex incubated at room temperature for 5 more minutes. Finally, the complex was allowed to reach equilibrium at room temperature for 10 min prior to use in crystallization experiments.

Crystals were grown at 19 °C in sitting drop trays in which 3 µL of ribosome complex was mixed with 4 µL of reservoir solution containing 100 mM Tris-HCl (pH 7.6), 150 mM L-Arginine-HCl, 2.9% (wt/vol) PEG 20,000, 9–10.5% (vol/vol) MPD, and 0.5 mM β-ME. Ribosome crystals grew to full size within 7–10 days. The crystals were transferred stepwise into cryo-protectant solutions with increasing MPD concentrations to 40% (vol/vol) and containing 100 mM Tris-HCl pH 7.6, 50 mM KCl, 10 mM NH$_4$Cl, 10 mM Mg(COOH)$_2$, 2.9% PEG 20,000, and 100 µM AMK or KAN in which they were incubated overnight at 19 °C. After stabilization, crystals were harvested and immediately frozen in a nitrogen cryostream at 80 K before being plunged into liquid nitrogen.

Collection and processing of the X-ray diffraction data, model building, and structure refinement were performed as described[33,64]. Diffraction data were collected at beamline 24-ID-C and 24-ID-E at the Advanced Photon Source at the Argonne National Laboratory (Argonne, IL) using NE-CAT remote access software 6.2.0. The final complete datasets of the 70S ribosome-AMK and -KAN complexes were both collected from a single crystal at 100 K with 0.3° oscillations and 0.979 Å wavelength. The raw data were integrated and scaled with the XDS program package (June 17, 2015)[65]. The ribosome complex with AMK or KAN, tRNA$^{Phe}$ and tRNA$_i^{fMet}$ crystallized in the primitive orthorhombic space group $P2_12_12_1$ with approximate unit cell dimensions 210 Å × 450 Å × 620 Å and contained two copies of the 70S ribosome per asymmetric unit of the crystal. The structure was solved by molecular replacement with PHASER from the CCP4 suite[66]. The search model was generated from the published high-resolution structure of the T. thermophilus 70S ribosome with all ligands removed. The initial molecular replacement solution was refined by rigid-body refinement with the ribosome split into multiple domains, followed by five cycles of positional and individual B-factor refinement with PHENIX 1.14[67]. After initial refinement, there was clear electron density in the unbiased $F_o − F_c$ difference Fourier maps corresponding to the mRNA, three tRNAs in the A, P, and E sites, and AMK or KAN.

Structural models and restraints for AMK and KAN were generated using PHENIX eLBOW[68]. The mRNA, AMK or KAN, tRNA$^{Phe}$ in the A site, tRNA$_i^{fMet}$ in the P site, and tRNA$^{Phe}$ in the E site were built into the unbiased difference density map from the initial round of refinement, and the refinement scheme described above was performed after addition of each ligand. The final model of the ribosome complex was generated by multiple rounds of model building in Coot 0.8.9.1[69] and subsequent refinement in Phenix 1.14[70]. The statistics of data collection and refinement for the complex are compiled in the Supplementary Table 2.

## Cryo-EM data acquisition, image processing, and structure determination

To determine the cryo-EM structure of the E. coli 70S-AMK complex with tRNAs, we incubated 2 µM E. coli 70S ribosomes purified from

strain MRE600 as described in ref. 64, 8 µM 24-MF mRNA, 8 µM fMet-tRNA$_i^{fMet}$ in 1x ribosome buffer (5 mM Tris-HCl pH 7.4, 60 mM NH$_4$Cl, 10 mM MgCl$_2$, 6 mM β-mercaptoethanol) at 37 °C for 10 min. Then, 40 µM AMK was added and the complex was incubated at room temperature for 10 min. Finally, 15 µM Phe-tRNA$^{Phe}$ was added and incubated for an additional 10 min at room temperature.

Quantifoil R2/1 gold 200 mesh grids (Electron Microscopy Sciences) were glow-discharged for 30 s in an (H$_2$O$_2$)-atmosphere using the Solarus 950 plasma cleaner (Gatan). Before freezing, the complex was diluted 1.5-fold in the 1x ribosome buffer, resulting in a final concentration of 1.3 µM 70S ribosomes and 25 µM AMK. The mixture (4 µL) was applied onto grids, blotted in 85% humidity at 22 °C for 24 s, and plunged-frozen in liquid nitrogen-cooled ethane using a Leica EM GP2 cryo-plunger. Grids were transferred into a Titan Krios G3i electron microscope (ThermoFisher Scientific) operating at 300 keV and equipped with a K3 direct electron detector camera (Gatan) mounted to a BioQuantum imaging filter operated with an energy filter slit width of 20 eV. Multi-shot multi-hole acquisition was performed by recording five shots per grid hole from nine holes at a time (3 × 3), using SerialEM[71] setup to record movies with 41 fractions with a total accumulated dose of 40.58 e$^-$/Å$^2$/movie. The nominal magnification was 105,000× and the pixel size at the specimen level was 0.839 Å. A total of 10,000 image stacks were collected with a defocus ranging between −0.7 and −2 µm. The statistics of data acquisition are summarized in Supplementary Table 3.

The image stacks (movies) were imported into cryoSPARC 4.1.2[72] and gain corrected. Image frames (fractions) were motion-corrected with dose-weighting using the patch motion correction, and patch contrast transfer function (CTF) estimation was performed on the motion-corrected micrographs. Based on relative ice thickness, CTF fit, length, and curvature of motion trajectories, 9,403 micrographs were selected for further processing (Supplementary Fig. 8).

1,981,432 particles were picked using the circular "blob" picker in cryoSPARC and were filtered based on defocus adjusted power and pick scores to 1,758,570 particles. Then, 1,435,519 particles were extracted (512 × 512-pixel box) and subjected to two rounds of reference-free two-dimensional (2D) classification. After discarding bad particles, 974,230 particles were selected from 2D classification and used to generate the ab-initio volume. Using 'heterogeneous refinement' in cryoSPARC with two groups, the 70S-like particles were further classified into one class average. This class represents the 70S ribosome with density for bound tRNAs. The particles (837,845) were binned 2x and were further classified based on focused 3D variability analysis (3DVA)[73] with a spherical mask around the acceptor stem of the P-site tRNA and AMK bound proximal to the CCA-end in the 50S subunit. This approach yielded to one main class containing 234,339 ribosome particles with solid density for the P-site tRNA and AMK bound to the 50S subunit P site. The particles were re-extracted to full-size (512×512-pixel box), followed by non-uniform and CTF refinement in cryoSPARC. The Fourier Shell Correlation (FSC) curves were calculated using the cryo-EM validation tool in Phenix 1.19.2 for even and odd particle half-sets masked with a 'soft mask' excluding solvent[74]. The E. coli 70S ribosome reconstruction complexed with AMK, mRNA, A-site Phe-tRNA$^{Phe}$, deacylated P-site tRNA$_i^{fMet}$, and deacylated E-site tRNA$^{Phe}$ has a nominal resolution of 2.9 Å using the FSC-cutoff criterion of 0.143 (Supplementary Fig. 9).

The previous E. coli 70S ribosome structure (PDB 8EKC)[64] was used to build the 70S-AMK complex. The individual 30S and 50S subunits were rigid-body docked into the 2.9 Å-resolution EM map using UCSF Chimera 1.14[75]. The Phe-tRNA$^{Phe}$ in the A site, tRNA$_i^{fMet}$ in the P site, the tRNA$^{Phe}$ in the E site were adjusted in Coot 0.9.8.7[69], and AMK was modeled bound to helix h44 and in the P site of the 50S subunit. The complete model of the E. coli 70S ribosome, including modified nucleotides in rRNAs and tRNAs, ordered solvent, bound AMK, A-site Phe-tRNA$^{Phe}$, P-site tRNA$_i^{fMet}$, and the E-site tRNA$^{Phe}$ was

real-space refined into the EM map for five cycles using Phenix 1.19.2[70] with global energy minimization and group ADP refinement strategies along with base pair restraints for rRNA and tRNAs, together with Ramachandran and secondary structure restraints. The resulting model of the E. coli 70S-AMK ribosome complex with Phe-tRNA$^{Phe}$ in the A site, deacylated tRNA$_i^{fMet}$ in the P site, and deacylated tRNA$^{Phe}$ in the E site was validated using the comprehensive validation tool for cryo-EM in Phenix 1.19.2[70]. The cryo-EM data collection, refinement, and validation statistics are compiled in the Supplementary Table 3.

### Inter-subunit contact surface area at bridge B5
The solvent accessible surface area at inter-subunit bridge B5 (SASA$_{B5}$) was calculated in PyMOL using the refined T. thermophilus 70S ribosome structure complexed with AMK. With the magnesium ions and water molecules removed, ribosomal elements and protein uL14 with generated hydrogen atoms within a radius of 15 Å around AMK were considered for the calculations. The interface area at bridge B5 without AMK is given by SASA$_{B5}$ = (SASA$_{B5(30S)}$ + SASA$_{B5(50S)}$) − SASA$_{B5(30S+50S)}$. Similarly, the interface area with AMK bound is given by SASA$_{B5+AMK}$ = (SASA$_{B5(30S)}$ + SASA$_{B5(50S+AMK)}$) − SASA$_{B5(30S+50S+AMK)}$.

### Figures
All figures showing electron density and atomic models were generated with PyMOL (The PyMOL Molecular Graphics System, Version 2.1.0 Schrödinger, LLC), the chemical structures of AMK and KAN were generated with the ChemDraw Professional version 16.0 software (PerkinElmer Informatics Inc.), and individual panels assembled with Adobe Illustrator (Adobe Inc.).

### Reporting summary
Further information on research design is available in the Nature Portfolio Reporting Summary linked to this article.

## Data availability
The data supporting the findings of this study are available from the corresponding authors upon reasonable request. The atomic coordinates and structure factors for the crystal structures of the T. thermophilus 70S ribosome complexes have been deposited in the Protein Data Bank (PDB) under the accession codes 8EV6 (70S-AMK) and 8EV7 (70S-KAN). The cryo-EM map of the E. coli 70S ribosome bound to AMK has been deposited in the Electron Microscopy Data Bank (EMDB) under the accession code EMD-40882, and the corresponding atomic coordinates in the PDB under the accession code 8SYL. Source data are provided with this paper.

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

## Acknowledgements

We thank members of the Sanyal and Gagnon laboratories for critical reading of the manuscript and suggestions. We are thankful to Y. Polikanov for advice with the generation of models and restraint files for AMK and KAN, and for sharing his custom-made Python scripts. This work was supported by a training fellowship from the Gulf Coast Consortia, on the Houston Area Molecular Biophysics Program (Grant No. T32-GM008280), by NIH grant R01GM136936 (to M.G.G.), Welch Foundation grants H-2032-20200401 and H-2032-20230405 (to M.G.G.), startup funds from The University of Texas Medical Branch (to M.G.G.), a Rising Science and Technology Acquisition and Retention Program award from the University of Texas system (to M.G.G.), a Pilot Grant from the Institute for Human Infections and Immunity at The University of Texas Medical Branch (to M.G.G.), and Swedish Research Council [2016-06264, 2018-05946, 2018-05498]; Knut and Alice Wallenberg Foundation [KAW 2017.0055]; Carl Trygger's Foundation [CTS 18:338, CTS 19:806]; Wenner-Gren Foundation [UPD2017:0238] to S.S., N.P.P. is supported by a doctoral scholarship from Sven och Lilly Lawskis fond för naturvetenskaplig forskning. This work is based upon research conducted at the Northeastern Collaborative Access Team beamlines, which are funded by the National Institute of General Medical Sciences from the National Institutes of Health (P30-GM124165). The Eiger 16 M detector on 24-ID-E is funded by a NIH-ORIP HEI grant (S10-OD021527). This research used resources of the Advanced Photon Source, a U.S. Department of Energy (DOE) Office of Science User Facility operated for the DOE Office of Science by Argonne National Laboratory under Contract No. DE-AC02-06CH11357. We also thank M. Sherman for help with cryo-EM data acquisition, the Sealy Center for Structural Biology and Molecular Biophysics of the University of Texas Medical Branch at Galveston for providing critical infrastructure and expertise, the Sealy and Smith Foundation for supporting the Sealy Center for Structural Biology at the University of Texas Medical Branch, and K.Y. Wong and J. Perkyns for computational support.

## Author contributions

S.M.S. and M.G.G. designed the project; S.M.S. prepared the samples for crystallization and cryo-EM analysis; S.M.S. and M.G.G. collected, and processed the X-ray and cryo-EM data; S.M.S. built the models and made the structure figures; N.P.P. performed all biochemical and MIC experiments; A.D.T. and X.G. performed tRNA translocation and peptide release assays using the A1408G ribosomes; N.P.P. and S.S. analyzed the kinetics data, and S.M.S. and M.G.G. wrote the manuscript with contributions from N.P.P. and S.S. All authors reviewed, edited, and approved the manuscript.

## Funding

## Competing interests

The authors declare no competing interests.
