## [Peer Review File · Nature Communications]

REVIEWER COMMENTS

Reviewer #1 (Remarks to the Author):

Aminoglycoside antibiotics primarily target translation by corrupting its accuracy through binding to a conserved site located in helix 44 (h44) in the vicinity of the peptidyl transferase centre (PTC). Aminoglycosides can have a pleotropic effect: they both bind to additional sites (e.g. one located in h69, Borovinskaya NSMB 2007) and compromise other activities of the ribosome such as translocation, ribosome recycling and peptide release Hirokawa et al. 2007, Parjuli et al. 2021). Understanding of the relative activities of aminoglycosides towards different sites and ability to inhibit different partial reactions of the translation cycle of interest since it could assist the future development of these drugs, e.g. development of variants that display lower off-target effects such as inhibition of mitotranslation.

In this collaborative study Sunyal (biochemistry using *E. coli* 70S) and Gagnon (X-ray crystallography using *T. thermophilus* 70S) labs characterised the mode of action of amikacin (AMK), using kanamycin (KAN) for comparison. The study dissected the molecular mechanisms of AMK-mediated inhibition of translation through binding to three sites that are located at i) the decoding center (inhibition of translation accuracy / translocation), ii) the P site (peptide release) and iii) the intersubunit bridge B5 (ribosome recycling). Biochemical results with AMR are very similar to these reported earlier by the Sunyal lab for another aminoglycoside, arbekacin (Parjuli et al. 2021), with an interesting difference of stronger inhibition of RF-mediated peptide release. The structural results represent a step forward compared to the cryo-EM structure of the *A. baumannii* ribosome in complex with AMK (Nicholson et al 2020): the *A. baumannii* structure was solved with vacant 70S, and only the primary h44 binding site was filled. Both Parjuli and Nicholson used AMK at 100 μ M. This suggests that the additional sites require programmed 70S to promote the efficient antibiotic binding, and, therefore, when possible, programmed ribosomal complexes should be used to study antibiotic binding by structural methods.

This is a nicely executed study that uses two complementary approaches. The manuscript is well-written, the figures are clear. However, in the light of earlier studies (such as Borovinskaya NSMB 2007, Hirokawa et al. 2007, Parjuli et al. 2021, Nicholson et al 2020 and others), the presented dataset does not necessarily provide dramatic conceptual advances in our understanding how aminoglycosides work. It is a well-executed careful study on an exceedingly well-researched topic. It is hard to discover something really new.

Specific comments:

Given that structural work is done with *T. thermophilus* 70S and biochemistry with *E. coli* ribosomes, maybe, it is worth providing a conservation analysis (sequence? Structure?) of the binding sites in the two species?

Is there a possibility to estimate the relative affinities of the three AMK sites? Does inhibition of release factor-mediated peptidyl-tRNA hydrolysis and ribosome happen in vivo? Is this a physiologically relevant mechanism? Comparative experiments with KAN are AMK comparisons could be helpful. Could a in vivo approach be informative in establishing which site is hit first, e.g. resistance mutations can point towards the relevant importance of the individual sites. What is the effect of the A1408G mutation in h44 that ablates the primary binding site on the MIC to KAN and AMK? If the mutant is fully resistant to AMK, then the primary site is clearly the only in vivo-relevant. Ideally one would use RiboSeq or similar approach to answer this question.

Reviewer #2 (Remarks to the Author):

Aminoglycosides are a clinically important class of antibiotics that inhibit translation by binding to the ribosome. Generally, these compounds are thought to block the translocation step of translation by preventing the movement of the A- and P- tRNA into the P- and E-sites. In the study of Seely and coworkers, structures of amikacin and kanamycin were determined on the *Thermus thermophilus* 70S ribosome with mRNAs and tRNAs at 2.9Å resolution. As expected, both amikacin and kanamycin bind within h44 on the 30S, where they are known to prevent translocation. The exciting finding of the study is that a second binding site is observed for amikacin, but not kanamycin, at the peptidyltransferase center of the 50S subunit, where it interacts with the CCA-end of the P-site tRNA. Complementary biochemical assays show that amikacin, but not kanamycin, inhibits peptidyl-tRNA hydrolysis by release factors and ribosome recycling by RRF and EF-G, suggesting that the additional binding site of amikacin at the PTC is responsible for this difference. Interestingly, peptide bond formation is not per se affected, highlighting the specific nature of the amikacin mediated inhibition as well as different mechanisms used for peptide bond formation and peptide release. Overall, the structure and biochemical experiments are technically well-performed and interpreted. The results are clearly presented and the manuscript written in a clear way. However, there is one main limitation of the study that the authors appear not to have addressed and surprisingly, not even commented upon, namely, whether the novel site for amikacin at the PTC has any physiological relevance.

Major points

1. Would one expect a lower MIC for AMK compared to KAN since AMK has an additional inhibitory mechanism? This seems like something that could be easily checked. Also how does the IC50 for in vitro translation inhibition compare between AMK and KAN? This would be another very direct measure.
2. Based on the biochemistry presented, it would seem that AMK inhibits in the low uM range, yet the complexes for structural analysis were made with 100uM of AMK (or KAN), which is orders of magnitude higher and therefore it is not surprising that many additional binding sites are observed, but whether they are contribute to the inhibitory activity is another point. For example, are there not mutations in the primary binding site that confer resistance to aminoglycosides such as AMK and KAN? The paper would be much more convincing if the authors could show that ribosomes bearing these mutations were resistant to KAN but not to AMK...this could even be done in vivo.

Additional points:

1. I find it totally inappropriate to not inform the reader in the introduction that a structure of amikacin already exists on a ribosome from a human pathogen at a higher resolution than reported by the authors on their medically-irrelevant bacterial species. Also to refer to the study as recent is stretching it – it was published in 2020 – already three years ago. As far as I can tell, the results presented here for amikacin binding to the decoding site are not in any way different from those presented before and therefore not particularly interesting. The relevant sections need to be re-written to reflect this.
2. Line 104-105: Why is it “remarkable” that additional binding sites for AMK and KAN are observed on the 70S ribosome when the authors put pounds of drug into their complexes? Also I do not think that these sites provide any “insight into their mechanism of action” – perhaps the one at the PTC for AMK but not any of the others.
3. Line 171. My understanding was that after peptide release, the deacylated P-tRNA oscillates between the P and E-sites on the 50S subunit. This then provides the opportunity for RRF to bind – not that RRF displaces the P-tRNA – it just prevents it moving back from the E-site.
4. It is nice that the authors took advantage of the EMPIAR data to reprocess the previous amikacin dataset, however, this should not be an aside but part of the paper i.e. inclusion in methods. I understand that the previous study used focused refinements and that the reprocessing was necessary to have a full 70S map to look at the interface. However, was amikacin and the CCA-end of the P-tRNA observed at the PTC site in the original 50S map? It sounds like the P-site tRNA was substoichiometric, therefore, since the authors re-processed the data, could they sort out a more defined P-tRNA-containing subpopulation with AMK? There is no description of how it was processed, which should be included. I would even go as far to say that authors could easily deposit a map and model for any homogenous P-tRNA-AMK-70S complex that they can sort and refine.

Reviewer #3 (Remarks to the Author):

Amikacin (AMK) is a clinically important aminoglycoside antibiotic that binds to the bacterial ribosome and is thought to exert its inhibitory effects by targeting multiple steps of the translation cycle. Understanding the mechanism of action of this drug is therefore a key step towards the design of new aminoglycosides to treat multidrug-resistant infections. In this work, Seely et al. dissect the pleiotropic effects exerted by AMK and its parent aminoglycoside compound, kanamycin (KAN), using a combination of in vitro fast kinetics and X-ray crystallography. While the kinetic experiments show that AMK inhibits the peptide release, recycling and translocation steps of translation, the structures of *T. thermophilus* 70S ribosomes in complex with AMK or KAN reveal the existence of multiple drug binding sites beyond the canonical aminoglycoside binding site at the decoding center. In particular, the binding of AMK, but not KAN, to a novel location near the P-loop of the peptidyl transferase center (PTC) is presented as the likely reason for AMK's greater ability to inhibit peptide release and ribosome recycling.

This study addresses an important mechanistic question and the results from it could help guide the design of new aminoglycosides. The structural and biochemical aspects of the work are performed to a high standard, and I have only minor comments in this regard, which I have listed at the end of this review.

On the other hand, I find that the way the structural data are used to interpret the biochemical data should be reevaluated by the authors. In particular, they should moderate their claim that the altered binding spectrum of AMK observed in the crystal structure is responsible for widening its mode of ribosome inhibition. As I will detail below, the available structural and biochemical data do not unequivocally support such a claim, which relies on the assumption that the conditions under which the structural work was performed directly mirror those of the in vitro fast kinetic studies.

If we first consider the concentration of the antibiotic, the kinetic experiments on RF2-mediated peptide release used concentrations of AMK up to 1 μM (Fig. 4a,b) whereas the structural work used 100 times this concentration (100 μM). The same point applies to the ribosome recycling experiments shown in Fig. 4c,d, where inhibition of ribosome recycling reaches a plateau around 10 μM AMK, a concentration 10-fold lower than that used for the crystal structure. Similarly, the effects of AMK on mRNA-tRNA movement during translocation appear to be maximal at a drug concentration of 2 μM (Fig. 2e,f).

If we turn our attention to the ribosome, the final concentration used for the fast kinetics experiments ranged from 0.05 μM (RF-mediated peptide release) to 0.25 μM (ribosomal translocation assay; recycling assay). In contrast, the concentration of ribosomes in a *T. thermophilus* 70S crystal is ~ 0.5 mM (calculated based on unit cell dimensions and the presence of 2 ribosomes per asymmetric unit).

In other words, the binding of multiple drug molecules to the ribosome observed in the crystal structures occurs at ribosome and drug concentrations that are 4 and 1-2 orders of magnitude higher, respectively, than in the biochemical experiments.

It should also be noted that a cryo-EM structure of the *A. baumannii* 70S ribosome in complex with AMK revealed a drug molecule bound to the same location in the PTC as in the crystal structure (Ref. 28, Extended Data Fig. 10). In this study, the ribosome concentration used was 120 nM (i.e. within the range used for the fast kinetics experiments in the present study), but the AMK concentration used was 100 μ M (same as in the crystal structure, i.e. 1-2 orders of magnitude higher than in the kinetics experiments). As a result, it is difficult to directly compare the structural results obtained with *A. baumannii* 70S with the kinetics data from this work.

In summary, one cannot conclude that the additional binding site for AMK observed at the PTC is occupied at the drug concentrations used for the in vitro fast kinetics experiments, and the inhibition of peptide release or ribosome recycling observed in these experiments could just as easily be due to a single AMK molecule binding near the decoding center. The greater affinity of AMK compared to KAN for the decoding center (resulting from the additional contacts made by its AHB moiety), could indeed account for their different effects on peptide release and recycling.

In the absence of structural data obtained under conditions similar to those of the biochemical experiments, I therefore recommend that the authors revise their manuscript to address this major point and tone down the conclusions relating to the additional AMK binding site near the PTC.

Specifically, the authors should:

- Explicitly mention in the main text of the manuscript the vastly different ribosome/drug concentrations used for the biochemical and structural parts of the work.
- Explain how these different ribosome/drug concentrations could lead to the observation of low affinity binding sites in the crystal structures. The current dismissal of some binding sites as low affinity on the basis that they are only observed in one of the ribosomes in the asymmetric unit is arbitrary, and all additional binding sites beyond the canonical aminoglycoside binding site should be treated as potentially low affinity unless there is evidence to the contrary.
- Remove any definitive assertion that the additional AMK molecule near the PTC is responsible for the observed pleiotropic effects for this drug. This hypothesis certainly deserves further testing, but the greater affinity of AMK for the decoding center would be an equally valid explanation for the observed

effects on peptide release and recycling. In my opinion, this simpler explanation should be favored and the authors should reinterpret their biochemical data in light of the additional contacts that AMK makes with the decoding center, rather than on the presence of the extra AMK molecule bound near the PTC.

- Reorganize the Results section to reduce the emphasis on the additional drug binding sites. In particular, I suggest that the authors not give these sites their own subsections, but rather combine them into a single subsection at the end of the Results section, where the alternative explanation relying on the AMK binding site at the PTC could be briefly presented in light of the caveats above.

In addition, the authors should address the following minor points:

- Line 99: The drug concentration should be specified. To allow better comparison with the kinetics experiments, the authors should provide both the concentration of the drug in the stabilization buffer and the concentration of ribosomes inside the crystal.

- The idea that the extra binding site near the PTC could be used to generate aminoglycosides with a shifted binding site is interesting and could be further developed in the discussion. As the authors are well aware, combining low-affinity binders to obtain a high-affinity binder is a well-established drug design strategy that could be applied to this case. Such a strategy could be implemented with AMK even if the additional binding site at the PTC turned out to be low affinity, so examples of how this could be achieved with reference to specific chemistries would be a welcome addition to the manuscript.

- Supplementary Table 1 - Please provide the clashscore and molprobit score for both structures.

REVIEWER COMMENTS

We are grateful to the reviewers for their insightful comments and suggestions. We believe that the revised version of the manuscript is much improved and addresses the reviewers' concerns. Our responses to the reviewers' inquiries are in blue.

Reviewer #1 (Remarks to the Author):

Aminoglycoside antibiotics primarily target translation by corrupting its accuracy through binding to a conserved site located in helix 44 (h44) in the vicinity of the peptidyl transferase centre (PTC). Aminoglycosides can have a pleiotropic effect: they both bind to additional sites (e.g. one located in h69, Borovinskaya NSMB 2007) and compromise other activities of the ribosome such as translocation, ribosome recycling and peptide release Hirokawa et al. 2007, Parjuli et al. 2021). Understanding of the relative activities of aminoglycosides towards different sites and ability to inhibit different partial reactions of the translation cycle of interest since it could assist the future development of these drugs, e.g. development of variants that display lower off-target effects such as inhibition of mitotranslation.

In this collaborative study Sunyal (biochemistry using *E. coli* 70S) and Gagnon (X-ray crystallography using *T. thermophilus* 70S) labs characterised the mode of action of amikacin (AMK), using kanamycin (KAN) for comparison. The study dissected the molecular mechanisms of AMK-mediated inhibition of translation through binding to three sites that are located at i) the decoding center (inhibition of translation accuracy / translocation), ii) the P site (peptide release) and iii) the intersubunit bridge B5 (ribosome recycling). Biochemical results with AMK are very similar to those reported earlier by the Sunyal lab for another aminoglycoside, arbekacin (Parjuli et al. 2021), with an interesting difference of stronger inhibition of RF-mediated peptide release. The structural results represent a step forward compared to the cryo-EM structure of the *A. baumannii* ribosome in complex with AMK (Nicholson et al 2020): the *A. baumannii* structure was solved with vacant 70S, and only the primary h44 binding site was filled. Both Parjuli and Nicholson used AMK at 100 μ M. This suggests that the additional sites require programmed 70S to promote the efficient antibiotic binding, and, therefore, when possible, programmed ribosomal complexes should be used to study antibiotic binding by structural methods.

This is a nicely executed study that uses two complementary approaches. The manuscript is well-written, the figures are clear. However, in the light of earlier studies (such as Borovinskaya NSMB 2007, Hirokawa *et al.* 2007, Parjuli *et al.* 2021, Nicholson *et al.* 2020 and others), the presented dataset does not necessarily provide dramatic conceptual advances in our understanding how aminoglycosides work. It is a well-executed careful study on an exceedingly well-researched topic. It is hard to discover something really new.

We are thankful to the reviewer for his comments and suggestions on our manuscript.

Specific comments:

Given that structural work is done with *T. thermophilus* 70S and biochemistry with *E. coli* ribosomes, maybe, it is worth providing a conservation analysis (sequence? Structure?) of the binding sites in the two species?

Response: This is a valid point and we thank the reviewer for bringing this to our attention. The primary aminoglycoside binding site in helix h44 is fully conserved between *T. thermophilus* and *E. coli* (see alignment below).

```
                1403  1409      1491  1498
                |    |      |    |
T.thermophilus  ccgucac----gaagucgu
E.coli          ccgucac----gaagucgu
                *      *      *      *
                *      *      *      *
```

The interactions between AMK and rRNA and tRNA at the second site near the P-site tRNA CCA-end are essentially mediated by the phosphate backbone. The nucleobase of G2252 in the P-loop is involved; however, this is a universally conserved nucleotide in the 50S subunit. The third site at the interface of the 30S and 50S subunits is not conserved. While the main interactions are mediated by the phosphate backbone of rRNA, AMK at this site also interacts with ribosomal protein uL14. The residues in uL14, Tyr7 and Glu54, interacting with AMK are not conserved between *E. coli* and *T. thermophilus*. This analysis is described in the results and discussion sections, and a sequence alignment of uL14 is shown in the Extended Data Fig. 11 to illustrate this point.

In the discussion:

Lines 333-339:

"We first analyzed the conservation of the residues in ribosomal protein uL14 at bridge B5 that interact with AMK. The most striking difference between *T. thermophilus* and *E. coli* is at position 54 (Extended Data Fig. 11). In *T. thermophilus*, Glu54 forms a hydrogen bond with the amine of ring III in AMK (Fig. 4c). In other representative bacteria, a basic residue (K or R) occupies position 54 (Extended Data Fig. 11), which may alter the binding site for AMK. This observation suggested that AMK may not bind to bridge B5 in the *E. coli* ribosomes, which we used in the kinetics experiments."

To further address the concerns of this reviewer, we now include a new cryo-EM structure of the *E. coli* 70S ribosome bound to AMK, mRNA and tRNAs. In agreement with the AMK-binding site conservation analysis, AMK is not observed between the subunits. However, clear density is seen at the primary AMK binding site in h44 as well as to the same site near the CCA-end of the P-site tRNA in the 50S subunit. The *E. coli* 70S-AMK complex was assembled using 1.3 μM 70S ribosomes and 25 μM AMK, which represents a ratio of 1:20 which is similar to that used in the kinetics assays.

In the results:

Lines 266-273:

"However, there is no density for AMK at bridge B5 in the *E. coli* ribosome, corroborating the non-conserved nature of Tyr7 and Glu54 in ribosomal protein uL14 (Extended Data Fig. 11), residues that interact with AMK at this site in the *T. thermophilus* ribosome (Fig. 4c). Increasing the concentration of AMK to 100 μM , the same as we used to soak the *T. thermophilus* ribosome crystals, did not populate this site (not shown), indicating that it is species-specific. Therefore, the inhibition of recycling of the *E. coli* ribosomes is not mediated through the binding of AMK at the subunit interface."

Is there a possibility to estimate the relative affinities of the three AMK sites? Does inhibition of release factor-mediated peptidyl-tRNA hydrolysis and ribosome happen in vivo? Is this a physiologically relevant mechanism? Comparative experiments with KAN are AMK comparisons could be helpful. Could a in vivo approach be informative in establishing which site is hit first, e.g. resistance mutations can point towards the relevant importance of the individual sites. What is the effect of the A1408G mutation in h44 that ablates the primary binding site on the MIC to KAN and AMK? If the mutant is fully resistant to AMK, then the primary site is clearly the only in vivo-relevant.

Response: The estimation of the relative affinity for the observed binding sites is not straightforward. However, in the light of the new experimental data we now provide, including the cryo-EM structure of the *E. coli* 70S-AMK complex and the kinetic analyses

of translocation and termination with A1408G mutant *E. coli* ribosomes, in which the canonical aminoglycoside binding site has been ablated, we can state that the primary binding site of AMK in h44 is the one responsible for the inhibitory effects we observed. The binding site in the 50S subunit P site does not seem to contribute to the inhibition by AMK. This is consistent with MIC measurement with WT and A1408G *E. coli* strains with AMK and KAN. For both antibiotics the MIC increases significantly with the A1408G mutation; MIC of KAN for WT (8 µg/mL) increases to (≥ 256 µg/mL) for A1408G. Similarly MIC of AMK for WT (1 µg/mL) increases to (≥ 16 µg/mL) for the A1408G strain. Furthermore, particle classification based on the presence of density near the CCA-end of the P-site tRNA shows that less than 30% of the ribosomes contain bound AMK at this site (see new Extended Data Fig. 8). This suggests that AMK binds to the 50S subunit P site with a lower affinity than the canonical site. This is mentioned in the discussion section:

Lines 347-352:

“However, three-dimensional (3D) variability analysis focused on the AMK binding site in the 50S subunit revealed that less than 30% of the ribosomes contained clear density for the drug at this site (Extended Data Fig. 8), suggesting that AMK has a lower affinity for the P site of the 50S subunit than for the canonical site in h44. This observation is also consistent with the 16-fold higher MIC of AMK for the *E. coli* strain expressing A1408G mutant ribosomes (Supplementary Table 1).”

Ideally one would use RiboSeq or similar approach to answer this question.

Response: We wholeheartedly agree with the review that obtaining ribosome profiling data for amikacin and kanamycin would be a valuable addition and expansion to this work. However, we believe these experiments would go beyond the scope of the current study, and we would like to remain focused here on the biochemical and structural aspects of AMK and KAN action.

Reviewer #2 (Remarks to the Author):

Aminoglycosides are a clinically important class of antibiotics that inhibit translation by binding to the ribosome. Generally, these compounds are thought to block the translocation step of translation by preventing the movement of the A- and P- tRNA into the P- and E-sites. In the study of Seely and coworkers, structures of amikacin and kanamycin were determined on the *Thermus thermophilus* 70S ribosome with mRNAs and tRNAs at 2.9Å resolution. As expected, both amikacin and kanamycin bind within h44 on the 30S, where they are known to prevent translocation. The exciting finding of the study is that a second binding site is observed for amikacin, but not kanamycin, at the peptidyltransferase center of the 50S subunit, where it interacts with the CCA-end of the P-site tRNA. Complementary biochemical assays show that amikacin, but not kanamycin, inhibits peptidyl-tRNA hydrolysis by release factors and ribosome recycling by RRF and EF-G, suggesting that the additional binding site of amikacin at the PTC is responsible for this difference. Interestingly, peptide bond formation is not per se affected, highlighting the specific nature of the amikacin mediated inhibition as well as different mechanisms used for peptide bond formation and peptide release. Overall, the structure and biochemical experiments are technically well-performed and interpreted. The results are clearly presented and the manuscript written in a clear way. However, there is one main limitation of the study that the authors appear not to have addressed and surprisingly, not even commented upon, namely, whether the novel site for amikacin at the PTC has any physiological relevance.

Response: We are thankful to this reviewer for pointing out a clear missing aspect of this work, the attribution of any physiological significance to the binding site of AMK in the P site of the 50S subunit. With the new structural, biochemical, and kinetics data we now provide, we attribute all the inhibitory effects of AMK to its binding to the canonical site in h44.

Major points

1. Would one expect a lower MIC for AMK compared to KAN since AMK has an additional inhibitory mechanism? This seems like something that could be easily checked. Also how does the IC50 for in vitro translation inhibition compare between AMK and KAN? This would be another very direct measure.

Response: We compared the minimum inhibitory concentrations (MICs) of AMK and KAN for WT and A1408G mutant strain of *E. coli* SQ171 using the standard broth microdilution method. Our results indicate that the MICs of KAN for WT (8 µg/mL) and

A1408G ($\geq 256 \mu\text{g/mL}$) were high compared to MICs of AMK for WT ($1 \mu\text{g/mL}$) and A1408G ($\geq 16 \mu\text{g/mL}$). The variation can be due to the additional AMK binding site at PTC, but can also be due to the difference in the affinity of these two drugs for the primary binding site at h44 of 16S rRNA within the 30S subunit. It is not possible to decipher this point from simple MIC measurement and comparisons.

The MIC values are now provided in Supplementary Table 1, and the above text is added in the results section (lines 300-305).

2. Based on the biochemistry presented, it would seem that AMK inhibits in the low μM range, yet the complexes for structural analysis were made with $100 \mu\text{M}$ of AMK (or KAN), which is orders of magnitude higher and therefore it is not surprising that many additional binding sites are observed, but whether they contribute to the inhibitory activity is another point. For example, are there not mutations in the primary binding site that confer resistance to aminoglycosides such as AMK and KAN? The paper would be much more convincing if the authors could show that ribosomes bearing these mutations were resistant to KAN but not to AMK...this could even be done *in vivo*.

Response: We have determined the cryo-EM structure of the *E. coli* 70S-AMK complex using $1.3 \mu\text{M}$ 70S ribosomes and $25 \mu\text{M}$ AMK, which represents a ratio of 1:20, similar to that used in the kinetics assays. This allowed to establish that AMK does not bind at the subunit interface (bridge B5) and therefore, no inhibitory effect can be attributed to this site. We observed that AMK still binds to the P site of the 50S subunit in the *E. coli* ribosome. To establish the relevance of this site, we performed kinetics with mutant *E. coli* ribosomes (A1408G) in which the canonical aminoglycoside binding site is ablated. These experiments, shown in the new Extended Data Fig. 12, show that AMK in the 50S subunit does not interfere with translation. This is in contrast to neomycin and tobramycin, for which inhibition of ribosome recycling was observed using the A1408G ribosomes, indicating that the binding site in helix H69 was responsible for this inhibition (Ref. 28: Ying L *et al.* RNA 2019). As mentioned above (see comments to reviewer #1), particle classification based on the presence of density near the CCA-end of the P-site tRNA for AMK shows that less than 30% of the ribosomes contain bound AMK at this site (see new Extended Data Fig. 8), suggesting that AMK binds to the 50S subunit P site with a lower affinity than to the canonical site. The presented experiments and structures systematically eliminate the potential contribution of each secondary site, attributing the measured inhibition to AMK binding to h44. The other secondary sites reported for AMK in the structure of the *T. thermophilus* 70S ribosome, obtained with higher concentrations of AMK, are not populated in the cryo-EM structure of the *E. coli*

ribosome.

Additional points:

1. I find it totally inappropriate to not inform the reader in the introduction that a structure of amikacin already exists on a ribosome from a human pathogen at a higher resolution than reported by the authors on their medically-irrelevant bacterial species. Also to refer to the study as recent is stretching it – it was published in 2020 – already three years ago. As far as I can tell, the results presented here for amikacin binding to the decoding site are not in any way different from those presented before and therefore not particularly interesting. The relevant sections need to be re-written to reflect this.

Response: Thank you for pointing this out. We now cite this study in the introduction (lines 83-85). We agree that the binding site in the decoding center was expected. Previous crystal structures of a fragment of h44 bound to AMK elucidated how it binds to the primary site (Ref 9: Kondo J *et al.* *Biochimie* 2006). In this regards, the cryo-EM structure of the *A. baumannii* 70S ribosomes with amikacin (Ref 14: Nicholson D *et al.* *Structure* 2020), despite being that of medically-relevant ribosomes, did not contain mRNA and tRNAs and as such, was not physiologically relevant. To no surprise, amikacin was bound to h44.

2. Line 104-105: Why is it “remarkable” that additional binding sites for AMK and KAN are observed on the 70S ribosome when the authors put pounds of drug into their complexes? Also I do not think that these sites provide any “insight into their mechanism of action” – perhaps the one at the PTC for AMK but not any of the others.

Response: We agree with the reviewer and have deleted the word “remarkable” as multiple binding sites have been observed with other aminoglycosides (Ref 12: Borovinskaya MA *et al.* *Nat. Struct. Mol. Biol.* 2007). We are not attempting to attribute any function to the other AMK (and KAN) binding sites observed in the *T. thermophilus* 70S ribosome. Three AMK binding sites appeared to be of potential interest, which is why we illustrated on that. With the addition of the new kinetic results with the A1408G ribosomes, the claims have been toned down.

It is indeed difficult to determine the “real” concentration of ribosomes and drug inside the crystal. Knowing the unit cell dimensions, the space group ($P2_12_12_1$), and the presence of two ribosomes per asymmetric unit, we can estimate the concentration of ribosomes inside the crystal. There are four asymmetric units per unit cell and therefore 8 ribosomes per unit cell of the crystal. From that and the volume of the unit cell ($V =$

210Å x 450Å x 620Å), the concentration of ribosomes inside the crystal is estimated to be ~0.2 mM. The local concentration of AMK upon soaking the crystals with 100 µM of AMK may also be vastly different. Therefore, we agree with this reviewer that the conditions in the crystal are not physiologically relevant.

We have re-written the manuscript to better reflect this caveat. In the revised version, we incorporate crystal and cryo-EM structures with kinetics assays with wild-type and A1408G mutant ribosomes. The cryo-EM dataset was collected with 1.3 µM *E. coli* 70S ribosomes, mRNA, tRNAs, and 25 µM AMK. This represents a ratio of 1:20, which is in-line with the 1:40 maximum ratio used in the kinetics assays. We observe clear density for AMK in the 50S subunit site near the CCA-end of the P-site tRNA at the exact same location as reported in the *T. thermophilus* 70S ribosome (see new Extended Data Fig. 10). However, as predicted from the AMK-binding site conservation, AMK is not bound at the interface between the subunits of the *E. coli* ribosome. The other secondary binding sites found in the *T. thermophilus* ribosome are likely artifacts caused by the high drug concentration used, as they are not observed in the *E. coli* ribosome at a lower concentration of AMK. We now address this issue in the text.

3. Line 171. My understanding was that after peptide release, the deacylated P-tRNA oscillates between the P and E-sites on the 50S subunit. This then provides the opportunity for RRF to bind – not that RRF displaces the P-tRNA – it just prevents it moving back from the E-site.

Response: After peptide release, the ribosome oscillates between the rotated and non-rotated conformations. Based on structural studies, RRF can bind to either the rotated (Dunkle JA *et al.* Science 2011) or the non-rotated ribosome (Ref 12: Borovinskaya MA *et al.* Nat. Struct. Mol. Biol. 2007 & Ref 34: Zhou D *et al.* Nat. Struct. Mol. Biol. 2020). Whether RRF can displace the P-site tRNA remains ambiguous; a prior study (Heurgue-Hamard V *et al.* 1998 EMBO J. 17; 808-816) shows that RRF, together with EF-G and RF3, stimulate peptidyl-tRNA release (drop off) from the ribosome. This suggested that RRF may displace peptidyl-tRNA from the ribosome. However, in the light of the new results showing that the AMK binding site in the 50S subunit near the tRNA does not inhibit recycling, we have removed this discussion from the manuscript.

4. It is nice that the authors took advantage of the EMPIAR data to reprocess the previous amikacin dataset, however, this should not be an aside but part of the paper i.e. inclusion in methods. I understand that the previous study used focused refinements

and that the reprocessing was necessary to have a full 70S map to look at the interface. However, was amikacin and the CCA-end of the P-tRNA observed at the PTC site in the original 50S map? It sounds like the P-site tRNA was substoichiometric, therefore, since the authors re-processed the data, could they sort out a more defined P-tRNA-containing subpopulation with AMK? There is no description of how it was processed, which should be included. I would even go as far to say that authors could easily deposit a map and model for any homogenous P-tRNA-AMK-70S complex that they can sort and refine.

Response: In the original 50S map of the *A. baumannii* ribosome (EMD-10809), weak residual density for amikacin near the PTC can be discerned. However, the sub-optimal sharpening of the map in this region combined with the low occupancy of AMK near the PTC and of the P-site tRNA in this structure, make it difficult to interpret the density. This likely explains why the study (Ref 14: Nicholson D et al. Structure 2020) did not report this binding site for AMK. After reprocessing the EM data deposited in the EMPIAR-10406, we applied a focused 3D classification scheme with a mask around the AMK binding site in the 50S subunit P site and the P-site tRNA CCA-end. Despite this, we could not improve the density for AMK and the tRNA. This is probably caused by the heterogeneity and the sub-stoichiometric occupancy of the P-site tRNA in the *A. baumannii* ribosome (EMD-10809), combined with the low number of particles in the dataset (~50,000) (Ref 14: Nicholson D et al. Structure 2020).

In the revised version of the manuscript, we do not include the reprocessed data of the *A. baumannii* ribosome (EMD-10809) (Ref 14: Nicholson D et al. Structure 2020). We believe it would confuse the reader and that the cryo-EM structure of the *E. coli* ribosome we now report adequately addresses the importance of the secondary binding sites. We describe data processing in the methods section, present the workflow in the Extended Data Fig. 8, the local resolution map and FSC curves in the Extended Data Fig. 9, and have deposited the *E. coli* 70S-AMK model in the PDB under the accession code 8SYL and the map in the EMDB (EMD-40882).

Reviewer #3 (Remarks to the Author):

Amikacin (AMK) is a clinically important aminoglycoside antibiotic that binds to the bacterial ribosome and is thought to exert its inhibitory effects by targeting multiple steps of the translation cycle. Understanding the mechanism of action of this drug is therefore a key step towards the design of new aminoglycosides to treat multidrug-resistant infections. In this work, Seely *et al.* dissect the pleiotropic effects exerted by AMK and its parent aminoglycoside compound, kanamycin (KAN), using a combination of *in vitro* fast kinetics and X-ray crystallography. While the kinetic experiments show that AMK inhibits the peptide release, recycling and translocation steps of translation, the structures of *T. thermophilus* 70S ribosomes in complex with AMK or KAN reveal the existence of multiple drug binding sites beyond the canonical aminoglycoside binding site at the decoding center. In particular, the binding of AMK, but not KAN, to a novel location near the P-loop of the peptidyl transferase center (PTC) is presented as the likely reason for AMK's greater ability to inhibit peptide release and ribosome recycling.

This study addresses an important mechanistic question and the results from it could help guide the design of new aminoglycosides. The structural and biochemical aspects of the work are performed to a high standard, and I have only minor comments in this regard, which I have listed at the end of this review.

On the other hand, I find that the way the structural data are used to interpret the biochemical data should be reevaluated by the authors. In particular, they should moderate their claim that the altered binding spectrum of AMK observed in the crystal structure is responsible for widening its mode of ribosome inhibition. As I will detail below, the available structural and biochemical data do not unequivocally support such a claim, which relies on the assumption that the conditions under which the structural work was performed directly mirror those of the *in vitro* fast kinetic studies.

If we first consider the concentration of the antibiotic, the kinetic experiments on RF2-mediated peptide release used concentrations of AMK up to 1 μM (Fig. 4a,b) whereas the structural work used 100 times this concentration (100 μM). The same point applies to the ribosome recycling experiments shown in Fig. 4c,d, where inhibition of ribosome recycling reaches a plateau around 10 μM AMK, a concentration 10-fold lower than that used for the crystal structure. Similarly, the effects of AMK on mRNA-tRNA movement during translocation appear to be maximal at a drug concentration of 2 μM (Fig. 2e,f).

If we turn our attention to the ribosome, the final concentration used for the fast kinetics experiments ranged from 0.05 μM (RF-mediated peptide release) to 0.25 μM (ribosomal translocation assay; recycling assay). In contrast, the concentration of ribosomes in a *T. thermophilus* 70S crystal is ~ 0.5 mM (calculated based on unit cell dimensions and the presence of 2 ribosomes per asymmetric unit).

In other words, the binding of multiple drug molecules to the ribosome observed in the crystal structures occurs at ribosome and drug concentrations that are 4 and 1-2 orders of magnitude higher, respectively, than in the biochemical experiments.

It should also be noted that a cryo-EM structure of the *A. baumannii* 70S ribosome in complex with AMK revealed a drug molecule bound to the same location in the PTC as in the crystal structure (Ref. 28, Extended Data Fig. 10). In this study, the ribosome concentration used was 120 nM (i.e. within the range used for the fast kinetics experiments in the present study), but the AMK concentration used was 100 μM (same as in the crystal structure, i.e. 1-2 orders of magnitude higher than in the kinetics experiments). As a result, it is difficult to directly compare the structural results obtained with *A. Baumannii* 70S with the kinetics data from this work.

In summary, one cannot conclude that the additional binding site for AMK observed at the PTC is occupied at the drug concentrations used for the in vitro fast kinetics experiments, and the inhibition of peptide release or ribosome recycling observed in these experiments could just as easily be due to a single AMK molecule binding near the decoding center. The greater affinity of AMK compared to KAN for the decoding center (resulting from the additional contacts made by its AHB moiety), could indeed account for their different effects on peptide release and recycling.

Response: This is an excellent point and we are thankful to this reviewer for pointing that out. Indeed, from the data presented in the original version of the manuscript, one could not determine the physiological relevance of the secondary binding sites. We now report the cryo-EM structure of the *E. coli* ribosome bound to AMK, mRNA and tRNAs. In this experiment, we used 1.3 μM ribosomes and 25 μM AMK, closely mimicking the conditions used in the kinetics experiments. Under these conditions, the canonical aminoglycoside binding site in h44 and the one proximal to the P-site tRNA in the 50S subunit are occupied by AMK. The kinetic assays performed with the A1408G ribosomes clearly show that the inhibitory effects on mRNA translocation, release factor-mediated peptide release, and ribosome recycling, are attributed to the site in h44 near the

decoding center. Our discussion has been accordingly modified.

In the absence of structural data obtained under conditions similar to those of the biochemical experiments, I therefore recommend that the authors revise their manuscript to address this major point and tone down the conclusions relating to the additional AMK binding site near the PTC.

Specifically, the authors should:

- Explicitly mention in the main text of the manuscript the vastly different ribosome/drug concentrations used for the biochemical and structural parts of the work.

Response: The new version of the manuscript addresses this issue. We now have a section in the results “Cryo-EM structure of the *E. coli* ribosome bound to amikacin” which highlights the caveats from the *T. thermophilus* crystal structures obtained with a high concentration of drug and mentions the high concentration of ribosomes inside the crystal (~0.2 mM), making the conditions non-physiological.

- Explain how these different ribosome/drug concentrations could lead to the observation of low affinity binding sites in the crystal structures. The current dismissal of some binding sites as low affinity on the basis that they are only observed in one of the ribosomes in the asymmetric unit is arbitrary, and all additional binding sites beyond the canonical aminoglycoside binding site should be treated as potentially low affinity unless there is evidence to the contrary.

Response: The additional AMK sites identified in the structure of the *T. thermophilus* ribosome are not populated in the *E. coli* ribosome. The only other AMK binding site that was worth exploring is the one in the 50S subunit P site, which is also present in the *E. coli* ribosome. However, as described in the discussion section, AMK likely binds with a lower affinity to the site near the PTC relative to that in h44.

Lines 347-352:

“However, three-dimensional (3D) variability analysis focused on the AMK binding site in the 50S subunit revealed that less than 30% of the ribosomes contained clear density for the drug at this site (Extended Data Fig. 8), suggesting that AMK has a lower affinity for the P site of the 50S subunit than for the canonical site in h44. This observation also

agrees with the 16-fold higher MIC of AMK for the *E. coli* strain expressing A1408G mutant ribosomes (Supplementary Table 1).”

- Remove any definitive assertion that the additional AMK molecule near the PTC is responsible for the observed pleiotropic effects for this drug. This hypothesis certainly deserves further testing, but the greater affinity of AMK for the decoding center would be an equally valid explanation for the observed effects on peptide release and recycling. In my opinion, this simpler explanation should be favored and the authors should reinterpret their biochemical data in light of the additional contacts that AMK makes with the decoding center, rather than on the presence of the extra AMK molecule bound near the PTC.

Response: In the revised version of the manuscript, the additional data reported show that the canonical binding site in h44 is indeed responsible for the pleiotropic effects of AMK on the ribosome. In light of the suggestions made by this and other reviewers, the data has been reinterpreted based on the additional experiments designed to systematically eliminate the contribution of the secondary binding sites.

- Reorganize the Results section to reduce the emphasis on the additional drug binding sites. In particular, I suggest that the authors not give these sites their own subsections, but rather combine them into a single subsection at the end of the Results section, where the alternative explanation relying on the AMK binding site at the PTC could be briefly presented in light of the caveats above.

Response: The Results section has been reorganized. However, two secondary binding sites were of particular interest based on their location and potential effect these could have on the function of the ribosome. One AMK binding site is at the inter-subunit bridge B5, and the other is near the PTC in the 50S subunit. We believe that these sites should be presented and then, guide the reader through the systematic analysis we used to establish the physiological importance of these sites. At the end, the major contribution to ribosome inhibition is from AMK binding to the canonical site near the decoding center.

In addition, the authors should address the following minor points:

- Line 99: The drug concentration should be specified. To allow better comparison with

the kinetics experiments, the authors should provide both the concentration of the drug in the stabilization buffer and the concentration of ribosomes inside the crystal.

Response: The drug concentration in the cryo-protection buffer has been added, and ribosome concentration inside the crystal, calculated at ~0.2 mM, is now included in lines 255-258.

- The idea that the extra binding site near the PTC could be used to generate aminoglycosides with a shifted binding site is interesting and could be further developed in the discussion. As the authors are well aware, combining low-affinity binders to obtain a high-affinity binder is a well-established drug design strategy that could be applied to this case. Such a strategy could be implemented with AMK even if the additional binding site at the PTC turned out to be low affinity, so examples of how this could be achieved with reference to specific chemistries would be a welcome addition to the manuscript.

Response: We are very thankful to this reviewer for this excellent suggestion. At the end of the Discussion, we have included one well-known example of how the adjacency of two drugs, linezolid and sparsomycin, has been used to design radezolid.

- Supplementary Table 1 - Please provide the clashscore and molprobit score for both structures.

Response: The clashscore and molprobit score are now included in this table, which is now Supplementary Table 2.

REVIEWERS' COMMENTS

Reviewer #1 (Remarks to the Author):

The authors have fully addressed my concerns. Addition of the new E. coli cryo-EM structure as well as of an MIC study was very helpful for bridging the structural (*T. thermophilus*) and biochemical (*E. coli*) parts of this work.

Reviewer #2 (Remarks to the Author):

The authors have now performed additional experiments suggesting that the secondary sites observed for amikacin are non-physiological. For me this removes a lot of novelty of the findings. As reviewer 1 said "It is a well-executed careful study on an exceedingly well-researched topic. It is hard to discover something really new." Therefore, I do not believe that the relatively modest new insights will be of general interest to the readers of Nat Comm and that such findings would be better placed in a more specialised journal.

Reviewer #3 (Remarks to the Author):

The additional cryo-EM structure and kinetic experiments with the A1408G mutant confirm the importance of the main AMK binding site over secondary sites. I am fully satisfied with the authors' response to the reviewers' comments and congratulate them for producing a greatly improved manuscript.